



# Extraordinary bloom of toxin-producing phytoplankton enhanced by strong retention in offshore continental shelf waters

Valeria Ana Guinder*[1], Urban Tillmann[2], Martin Rivarossa[3], Carola Ferronato[1], Fernando J. Ramírez[1], Bernd Krock[2], Haifeng Gu[4], Martin Saraceno[5,6,7]

[1] Instituto Argentino de Oceanografía (IADO), Consejo Nacional de Investigaciones Científicas y Técnicas (CONICET), Argentina. Universidad Nacional del Sur, UNS. Florida 4750, 8000 Bahía Blanca, Argentina.

[2] Alfred Wegener Institut-Helmholtz Zentrum für Polar- und Meeresforschung, Am Handelshafen 12, 27570 Bremerhaven, Germany.

[3] Instituto Nacional de Investigación y Desarrollo Pesquero, INIDEP, and CONICET. Paseo Victoria Ocampo, Escollera Norte 1, B7602HSA Mar del Plata, Argentina.

[4] Third Institute of Oceanography, Ministry of Natural Resources, Xiamen 361005, China.

[5] Centro de Investigaciones del Mar y de la Atmósfera (CIMA), CONICET, Facultad de Ciencias Exactas y Naturales, Universidad de Buenos Aires (UBA), C1063ACV, Buenos Aires, Argentina.

[6] Instituto Franco-Argentino de Estudio sobre el Clima y sus Impactos (IFAECI) CNRS-IRD-UBA-CONICET, Buenos Aires.

[7] Departamento de Ciencias del Océano y de la Atmósfera (DCAO), Facultad de Ciencias Exactas y Naturales, Universidad de Buenos Aires (UBA), C1063ACV, Buenos Aires, Argentina.

*Correspondence to: vguinder@iado-conicet.gob.ar

**Abstract:**

The extensive Patagonian continental shelf in the Atlantic Ocean is renowned for its high productivity associated with nutrient-rich waters that fertilise massive phytoplankton blooms, especially along the shelf-break frontal system. Growing evidence reflects this ecosystem as a hotspot for harmful algal blooms (HABs). Whether these HABs reach coastal areas or are exported to the adjacent ocean basin by energetic edge currents remains unexplored. During two oceanographic cruises in spring 2021, a bloom of dinoflagellates of the Amphidomataceae family was sampled over the outer shelf with a ten-day interval, at stations 40 km apart. The bloom was first sampled on November 16, with 32 million cells L$^{-1}$, and was still persistent on November 25, with 14 million cells L$^{-1}$. The magnitude of this bloom is a global record for this group so far reported in the literature. The toxin azaspiracid-2 was detected in both stages of the bloom, with values up to 2122 pg L$^{-1}$. The most likely source of AZA-2 was *Azadinium spinosum* ribotype B. The bloom developed in vertically stable waters (60 m mixed layer depth) with elevated chlorophyll concentration. Water retention and the presence of fronts induced by horizontal stirring controlled the persistence and trajectory of the bloom in a localised area over the continental shelf, as evidenced by analysis of geostrophic surface currents, Lyapunov coefficients, and particle advection modelling. These findings underscore the importance of monitoring HABs in offshore environments, and the need to understand bio-physical interactions that govern bloom taxa assemblages and transport pathways.



## 1. Introduction

In marine environments, dinoflagellates are the primary toxin-producing group of protistan plankton and key causative agents of harmful algal blooms (HABs). As the most diverse group of toxic microorganisms, -e.g. *Alexandrium* spp., *Karenia* spp., *Dinophysis* spp., *Azadinium* spp., *Amphidoma* spp.-, dinoflagellates produce a wide range of toxins. Phycotoxins are natural intracellular metabolites synthesised by certain microalgae that can be transferred through the food web, having severe impacts on marine biota, ecosystems and human health (Anderson et al., 2015; Sunesen et al., 2021). In the Argentine Sea, records of HABs caused by different plankton species have risen since the first documentation of human poisoning in spring 1980 caused by paralytic shellfish toxins (PST), produced by *Alexandrium catenella* in coastal areas (reviewed in Ramírez et al., 2022). Broadly, HABs were long thought to occur exclusively in coastal regions, due to their visible impacts on water quality and human-related activities, as documented for instance in the Argentine Patagonian Gulfs (Wilson et al., 2015; D'Agostino et al., 2019) and the Beagle Channel (Almandoz et al., 2019; Cadaillon et al., 2024). However, the perception of HABs as solely coastal events was biased, primarily due to greater monitoring efforts in coastal areas compared to the fewer studies conducted offshore (Hallegraeff et al., 2021; Sunesen et al., 2021; Anderson et al., 2021). In line with this trend, the expansion of the monitored area over recent decades have confirmed that toxic species are indeed common in offshore waters in the Argentine Sea (Ramírez et al., 2022), especially in the outer continental shelf associated to the shelf-break front (reviewed in Guinder et al., 2024). Furthermore, the increase in oceanographic studies focused on detecting HABs along the outer Patagonian shelf has led to several new records of toxin-producing species and phycotoxins in the South Atlantic (Akselman et al., 2015; Guinder et al., 2018; Tillmann et al., 2019). In particular, large HABs formed by the nano-dinoflagellates of the Amphidomataceae family —producers of the toxin azaspiracids— have emerged as important hazards in the productive Patagonian shelf-break frontal ecosystem (Guinder et al., 2024). It is well known that dinoflagellates possess advantageous strategies for thriving in frontal systems, such as effective swimming, mucus and cyst formation, mixotrophy, and toxin production (Smayda, 2002; Glibert, 2016). However, the bio-physical mechanisms explaining the development of large harmful blooms on hydrographically complex shelves are still not fully understood mainly due to the lack of simultaneous taxonomic data and velocity fields at synoptic scales.

The extensive Patagonian shelf-break front (35-55°S) in the SW Atlantic Ocean is a high productivity ecosystem, located ~200 to ~900 km offshore (Martinetto et al., 2019; Guinder et al., 2024). This permanent termohaline front is associated with the upwelling of nutrient-rich waters of the westerly edge of the Malvinas Current, which fertilises the surface waters over the shelf (Palma et al., 2008; Matano et al., 2010). Additionally, over the mid shelf, the Patagonian Current transports diluted subantarctic waters northwards, also loading with nutrients the region. Hence, massive phytoplankton proliferations occur over the shelf in spring and summer (García et al., 2008; Carreto et al., 2016; Ferronato et al., 2023), including a variety of HAB-forming taxa and associated phycotoxins. The most conspicuous HABs are those formed by the dinoflagellates of the clade Amphidomataceae (Akselman and Negri, 2012; Fabro et al., 2019; Guinder et al., 2020; Tillmann et al., 2018; 2019), which include the four azaspiracids (AZAs)-producing species, i.e. *Azadinium dexteroporum*, *Az. poporum*, *Az. spinosum*, and *Amphidoma languida* (Krock et al. 2019). Amphidomataceans have been reported from different marine regions globally (Tillmann, 2018; Salas et al., 2021; Liu et al., 2023) but so far, the maximum bloom abundances reported in the literature are from the Argentine Sea (Akselman and Negri, 2012). During the springs of 1991 and 1992, these dinoflagellates reached between 3 and 9 million cells per litre and caused water discoloration in the northern area (38-42°S; 58-56°W) of the Patagonian shelf (Akselman and Negri, 2012). No toxin screening was performed at that time, but in spring 2015, the production of AZAs was confirmed in another large bloom in the area (Tillmann et al., 2019). Moreover, AZAs have been detected in the tissue of the scallop *Zygochlamys patagonica* since the early 90ies (Turner and Goya, 2015). These scallops form large seabed banks along the 100-metre isobath between 38°S-48°S (Alemany et al., 2024), associated with the high phytoplankton productivity over the outer Patagonian continental shelf.





Despite the limited synoptic sampling in offshore waters, the prevalence of HABs in the Patagonian front highlights this
ecosystem as a hotspot that requires further monitoring. The notably high abundance of Amphidomataceans over the outer
shelf holds greater significance when assessing the potential risks posed to both regional and global ecosystems. The evolution
and transport of HABs remain poorly understood, as does the question of whether they may reach coastal areas or be exported
offshore into the stirring Atlantic Ocean. In this study, we characterised the bio-physical aspects of a large multispecific spring
bloom of Amphidomataceans, detected through an unusual sampling effort that involved two research expeditions in
November 2021. This HAB was observed at two sampling sites 40 km apart within a span of 10 days. In oceanic waters, the
permanence and spatial extent of discrete phytoplankton blooms are influenced by dispersal mechanisms that rely on diffusion
and horizontal advection (Abraham et al., 2000; Mahadevan, 2016; Lehahn et al., 2017). Typically, the dispersion and stirring
of phytoplankton blooms in the ocean have been studied using remote sensing of chlorophyll-*a* and models (Lehahn et al.,
2007; Lévy et al., 2018; Ser-Giacomi et al., 2023), with few studies considering *in situ* sampling (Abraham et al., 2000;
Giddings et al., 2014; Hernández-Carrasco et al., 2020) to assess the bio-physical couplings of bloom development. Aside
from the key role of mesoscale energetic variability in modulating phytoplankton community, observational studies combining
multiple approaches at synoptic scale are still scarce. Here, we combined field observations of protistan plankton species
composition and associated toxins, with remotely sensed ocean colour images of chlorophyll-*a* and geostrophic surface
currents, particle tracking experiments, and Lyapunov coefficient analysis to assess the horizontal displacement and retention
of the Amphidomataceae bloom within a mesoscale eddy. Furthermore, we aim to explore whether this HAB that developed
in offshore shelf waters might reach coastal areas or be advected by the Malvinas Current, facilitating the dispersal of toxic
species to other shelves and ocean basins.
**2.   Materials and Methods**
**2.1 Hydrography and productivity in the Patagonian continental shelf**
Along the external margin of the Patagonian continental shelf, between 35°S and 55°S, a thermohaline front develops
throughout the year, characterised by high biological productivity. The development of this front and the associated upwelling
is due to the interaction of the energetic western boundary current system with the steep slope and the waters of the shelf, as
well as the effect of winds and tides (reviewed in Piola et al., 2024). The Malvinas Current originates at ~55°S as a branch of
the Antarctic Circumpolar Current **(Fig. 1)** and runs northwards at high velocity (mean surface velocities 45 cm/s, Piola et al.,
2024) along the shelf break in two jets that meet at ~44°S (Frey et al., 2023). Then, at ~38°S, the Malvinas Current meets the
warm and oligotrophic Brazil Current, which runs southwards, in the so-called Brazil-Malvinas Confluence, and waters are
exported eastwards into the South Atlantic Ocean basin **(Fig. 1)**. In addition, another branch of the Antarctic Circumpolar
Current gives origin to the Patagonian Current which runs northwards over the continental shelf, carrying diluted subantarctic
waters **(Fig. 1).**
The interaction of the Malvinas Current with the irregular bottom topography generates upwelling of cold, nutrient-rich waters
that fertilise phytoplankton over the shelf, together with the Patagonian Current (reviewed in Guinder et al., 2024).
Phytoplankton blooms expand over the mid and outer shelf as reflected by a persistent satellite chlorophyll-*a* band, wider in
spring and narrower in summer along the shelf-beak front (Guinder et al., 2024). The magnitude of the upwelling has low
seasonal variability and is heterogeneous along the extensive latitudinal range of the slope (Combes and Matano, 2018). Hence,
productivity over the shelf varies spatially and temporally, and in consequence, multiple bioregions emerge, each characterised
by unique phytoplankton phenological patterns, as revealed by climatological analysis of satellite-derived chlorophyll *a*
(Delgado et al., 2023).



**2.2 Research cruises**

Two oceanographic expeditions were carried out during late-spring (November 2021) to study the microbial plankton
communities in the Argentine continental shelf and adjacent ocean basin (**Fig. 1**). The first research cruise was the Ana María
Gayoso (hereafter Gayoso, GA) onboard the R/V Bernardo Houssay (PNA and CONICET, Arg.), which covered a sampling
period between 16 and 22 November 2021, at 10 stations along the outer Patagonian shelf, the core of the Malvinas Current,
and adjacent open ocean waters (**Fig. 1**). The second cruise was the Agujero Azul (AA) onboard the R/V Victor Angelescu
(INIDEP and CONICET, Arg.) which covered a sampling period between 25 November and 3 December 2021, at 23 stations
aligned in two cross-shelf transects in the so-called Agujero Azul area (44-47°S; 62-57.5°W) (**Fig. 1**). We first provide a
general overview of the sampling conducted in both cruises, but then we focus on the Amphidomataceae bloom observed over
the shelf at station GA01 on 16 November and at station AA09 on 25 November, separated by *ca.* 40 km (**Fig. 1**).

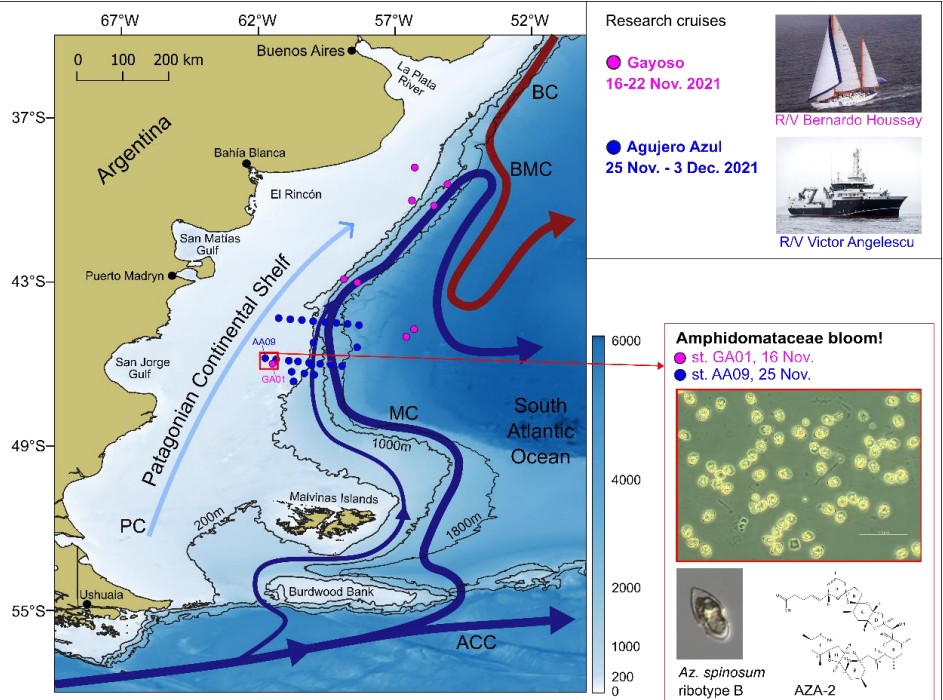


**Figure 1: Sampling stations of the research cruises Gayoso (pink dots, onboard R/V Houssay) and Agujero Azul (blue**
**dots, onboard R/V Angelescu), and the main circulation pattern in the Argentine continental shelf and shelf break. The**
**stations where the bloom of Amphidomataceae was observed are indicated with a red square: GA01 and AA09, sampled**
**on November 16 and 25, respectively. Micrographs show the Amphidomatacean bloom and a cell of *Azadinium***
***spinosum* ribotype B, producer of azaspiracid-2 (AZA-2). ACC: Antarctic Circumpolar Current, MC: Malvinas**
**Current, BC: Brazil Current, BMC: Brazil-Malvinas Confluence and PC: Patagonian Current. Bathymetry from**
**GEBCO, 2021. Isobaths of 200 m, 1000 m and 1800 m are displayed.**

**2.3 Remote sensing of surface Chl-*a*, SST and ADT**

In order to contextualise the discrete observations of plankton within broader spatio-temporal dynamics, we explored satellite-
derived surface chlorophyll-a concentration (Chl-a sat) data from the days of the cruises. In this sense, daily merged product



with a 4 km resolution, provided by the GlobColour project (distributed by ACRI ST, France: https://hermes.acri.fr/), were
downloaded. This ocean color product is generated from the fusion of the SeaWiFS, MERIS, MODIS-Aqua, and OLCI sensors
and estimates the average Chl-*a* concentration in the surface layer (Maritorena et al., 2010). The fusion of data from different
satellite sensors, combined with the quality control criteria used by GlobColour, enables enhanced spatial and temporal
coverage. Eight-day temporal averages were calculated for the periods of November 12–19 and November 20–27, 2021.
Additionally, daily Chl-a images were assessed from November 14 to 27 focused on the bloom area to track its short-term
evolution. Due to high cloud covering, data is missing from 22 to 24 November. To analyse Sea Surface Temperature (SST)
during the sampling periods, daily NSST MODIS-Aqua level L3 images with a 4 km resolution were downloaded from
https://oceancolor.gsfc.nasa.gov/. Eight-day temporal averages were also constructed for the periods of November 12–19 and
November 20–27, 2021. All images were processed using SeaDAS v8.3 and QGIS 3.38, mapped to the WGS84 reference
system (datum WGS84, ellipsoid WGS84), and restricted to the study area. The images were smoothed using a 'Non-Linear
Mean 3x3' filter.
To compute trajectories of virtual particles (see Sect. 2.6 below) and Finite Size Lyapunov Exponents (FSLE, Sect. 2.7) we
used Absolute Dynamic Topography (ADT) maps and geostrophic velocities derived from the ADT maps. Gridded ADT maps
of daily temporal resolution and ¼ of degree spatial resolution maps were obtained from CMEMS
(https://marine.copernicus.eu/). FSLE images with a spatial resolution of 1/25° grid were downloaded from AVISO
(https://www.aviso.altimetry.fr).

**2.4 *In situ* measurements and sample collection**

At each sampling station, continuous vertical profiles of temperature, salinity and fluorescence were measured. In Gayoso
cruise, a Sea-Bird 9 plus CTD and a fluorometer sensor Wet Labs FLRTD-5105 were used. In Agujero Azul cruise, a Seabird
SBE 9 Digiquartz CTD coupled with an ancillary Seapoint SCF chlorophyll fluorometer were used. To assess the vertical
stability of the water column, the Brunt-Väisäla buoyancy frequency (cyc h$^{-1}$) was computed using the function swN2 of the
package oce (Kelley et al. 2022) in R statistical software. Thereafter, the mixed layer depth (MLD, in metres) was defined at
the depth where the maximum value of the Brunt-Väisäla frequency was detected.
Niskin bottles attached to the CTD-rosette were used to collect water samples at the surface (5 m depth) for the analysis of
chlorophyll *a,* dissolved inorganic macronutrients, protistan plankton by microscopy, genetic analysis of the species diversity,
and phycotoxins in field samples.
For chlorophyll *a*, a volume of 400 mL was filtered through filter GF/F fiber-glass filters pre-combusted at 450 °C for 4 h. A
volume of 10 mL of 90 % acetone was used for pigment extraction during 24 hs (4 °C), and thereafter quantified using an
Agilent Cary 60 UV-Vis spectrophotometer. Concentration was estimated using the equations developed by Jeffrey and
Humphrey (1975).
For inorganic nutrients, the water samples filtered through Whatman GF/F fiber-glass filters pre-combusted at 450 °C for 4 h,
were stored at -20 °C in alkali-rinsed (NaOH, 0.1 M) polyethylene bottles. Nitrite, nitrate, ammonium, silicate and phosphate
were measured using a spectrophotometer Agilent Cary 60 UV-Vis following the method outlined by Hansen and Koroleff

189   (1999).

Duplicate samples for plankton counts collected with Niskin bottles were preserved with Lugol (1% f/c) and formaldehyde
(1% f/c) in glass bottles (250 mL) and kept in dark and at 4 °C for their analyses under microscopy. Similarly, duplicate water
samples were collected by three vertical net tows (20 µm size pore) integrating the first 30 m depth for the identification of
protists' taxa.
For the quantification of azaspiracids (AZAs) as well as for genetic analysis of field species diversity, the same sampling
protocol was applied. A volume of 4-5 L of seawater from the Niskin bottles was pre-screened through a 20 µm mesh-size,



and subsequently filtered through 5 µm pore-size polycarbonate filters (Millipore, Eschborn, Germany) under gentle vacuum
(< 200 mbar). Filters were placed in 50 mL centrifuge tubes and preserved at -80 °C for further analyses in the laboratory.

**2.5  Microscopy analysis of protistan plankton**

Morphological aspects of plankton cells were carefully observed under different light microscopes all equipped with
epifluorescence and differential interference contrast optics: a Nikon Eclipse E-400 microscope, a Zeiss Axioskop 2
microscope, and an inverted Axiovert 200 M. In order to measure length and width of cells, micrographs were taken at 1000
magnification under a Zeiss Axio Vert.A1 equipped with a digital camera AxioCam 208 Color, and under an Axioskop 2
equipped with a Axiovision digital camera. Thereafter, they were processed with the software ZEN (v.2.7, Zeiss) and
Axiovision (v.4.8, Zeiss). Further, scanning electron microscopy (SEM, FEI Quanta FEG 200) was used to assess detailed
taxonomic features of the dinoflagellate species (e.g. arrangement of thecal plates, presence of pores and spines, etc). SEM
samples were treated following the protocol described in Tillmann et al. (2017). For the estimation of total protists' abundance
(in cells $L^{-1}$), seawater samples collected with Niskin bottles and fixed with Lugol were settled in sedimentation chambers and
single cells were counted under inverted microscope using a magnification of 400 following traditional techniques (Hasle,
1987). All protists larger than 5 µm in cell size were counted and classified into species or genera taxonomic levels, or merged
into taxonomic/functional groups organized in size ranges (e.g. ciliates between 10-20 µm, cryptophytes <10 µm,
*Gymnodinium*-type cell, Kareniaceae-type cell, etc.). In addition, to assess the relative abundance of the Amphidomataceae
species responsible for the multispecific bloom of this clade, subsamples (10-mL) were carefully counted with high taxa
resolution.

**2.6. Genetic analysis**

For a broad detection of amphidomatacean species diversity in field samples, metarbarcoding was performed specifically
targeting the internal transcribed spacer (ITS1) region, following Liu et al. (2023). This information was used as a complement
to the exhaustive morphological taxonomy performed under light microscopy and SEM, especially for the accurate
identification of ribotypes.

**2.7 Toxin identification and quantification**

Filters were repeatedly rinsed with 500 µL methanol until complete discoloration of the filters. The methanolic extracts were
transferred to a spin-filter (0.45 µm pore-size, Millipore) and centrifuged at 800 × g for 30 s, followed by transfer to
autosampler vials and stored at −20 °C until analysis. Toxin analyses were performed using high performance liquid
chromatography coupled to tandem mass spectrometry HPLC-MS/MS in the selected reaction monitoring (SRM) mode for
the detection of known AZA variants. In addition, precursor experiments of the ions $m/z$ 348, 350, 360, 362, and 378 were
carried out to find potentially new AZA variants. Screened mass transitions and instrument parameters are detailed in Tillmann
et al. (2021).

**2.6  Lagrangian analysis**

We used the geostrophic currents computed from satellite altimeter data (see Sect. 2.3) and an algorithm that represents the
advection process caused by those currents to represent the trajectories of virtual neutrally buoyant particles. Particles were
released at the surface along 46°S every 0.05 grades and in the four regions indicated in the **Appendix D**. The algorithm used





for the advection process is fully described in Haller and Beron-Vera (2012). The algorithm computes the particle positions
based on initial location and knowledge of the velocity field. Therefore, the accuracy of the trajectories obtained relies on the
accuracy of the geostrophic velocity field obtained from satellite altimetry. In the northern portion of the Argentine continental
shelf, such surface velocities showed to be well correlated with in situ current measurements (Lago et al., 2021). We therefore
assume that the surface dynamics can be represented by satellite altimetry derived data and use it as the input velocity field for
the algorithm to advect the virtual particles.

**2.7  Lyapunov coefficient analysis**

In order to examine meso- and submesoscale frontal structures during phytoplankton blooms, daily finite-size Lyapunov
exponent (FSLE) images from November 10 to 25 were analysed. Additionally, the daily images were used to create a video
with Filmora v.11 (available in the **Appendix D**) to illustrate the daily evolution of the FSLE in the area where the
phytoplankton bloom developed. The FSLE is obtained by measuring the backward-in-time divergence of initially nearby
particles and it is commonly used as an indicator of frontal activity and stirring intensity. Relatively large FSLE values are
associated with formerly distant water masses, whose confluence creates a transport front (d'Ovidio et al., 2004; d'Ovidio et
al., 2009). Fronts identified as maxima (ridges) of FSLEs have a convergent dynamics transverse to them, so that passive
particles in their neighbourhood are attracted to the front and then advected along it (Della Penna et al., 2015).

**3.  Results**

**3.1 Satellite-derived chlorophyll *a* during the sampling period**

During the Gayoso cruise (12-19 Nov, **Fig. 2a**), a uniform, large band of high surface Chl-*a* concentration expanded over the
mid and outer shelf. During the Agujero Azul cruise (20-27 Nov, **Fig. 2b**), the band of Chl-*a* disaggregated and showed lower
intensity, but the Chl-*a* concentration was still high in the area of the sampling stations **(Fig. 2b)**. The distribution of Chl-*a*
concentration during the time of the cruises is indicative of mid-late spring phytoplankton bloom (November) over the
Patagonian shelf which typically exhibits higher concentration south of 43°S following the thermal stratification (**Fig. 2a, b**).
The SST showed warming of the inner-mid shelf waters north of 44°S over the 8-day average periods **(Fig. 2c, d),** but the SST
remained yet constant at the sampling stations' area (see **Fig. 4** and **Table 1**). The daily images of Chl-*a* from 14 to 25
November (**Fig. 3**) showed an abrupt proliferation of phytoplankton on November 15 which notably intensified on November
16 when it reached the maximum concentrations during the studied period. On this date, the extraordinary bloom of
Amphidomataceae was sampled at GA01. During the following days, the chlorophyll levels remained high in the area but
became more disaggregated into variable patches. On November 25, the sampling day at station AA09, Chl-*a* had decreased
at station GA01 but remained intense at AA09 with still extraordinary densities of Amphidomataceans.











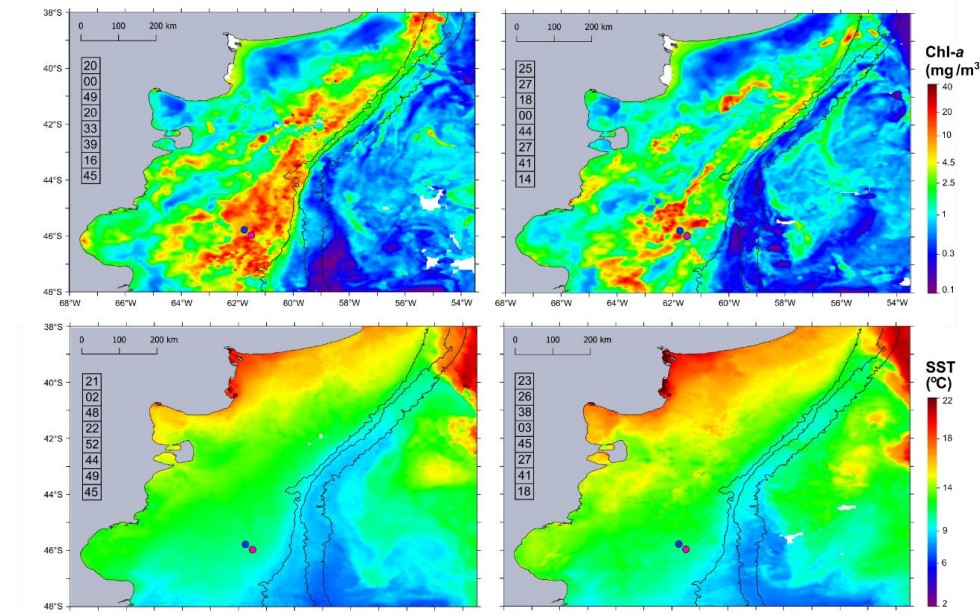

**Figure 2: Eight-day time mean of (a, b) satellite chlorophyll *a* (Chl-*a* in mg/m3) and (c, d) sea surface temperature (SST in °C) during the sampling period: (a and c) 12-19 November 2021, and (b and d) 20-27 November 2021. From left to right, isobaths of 200, 1000 m and 1800 m are shown. The numbers in the column on the left side of each panel correspond to the percentage of cloud-free pixels in each daily satellite image. The sampling stations GA01 (pink dot) and AA09 (blue dot) are shown.**

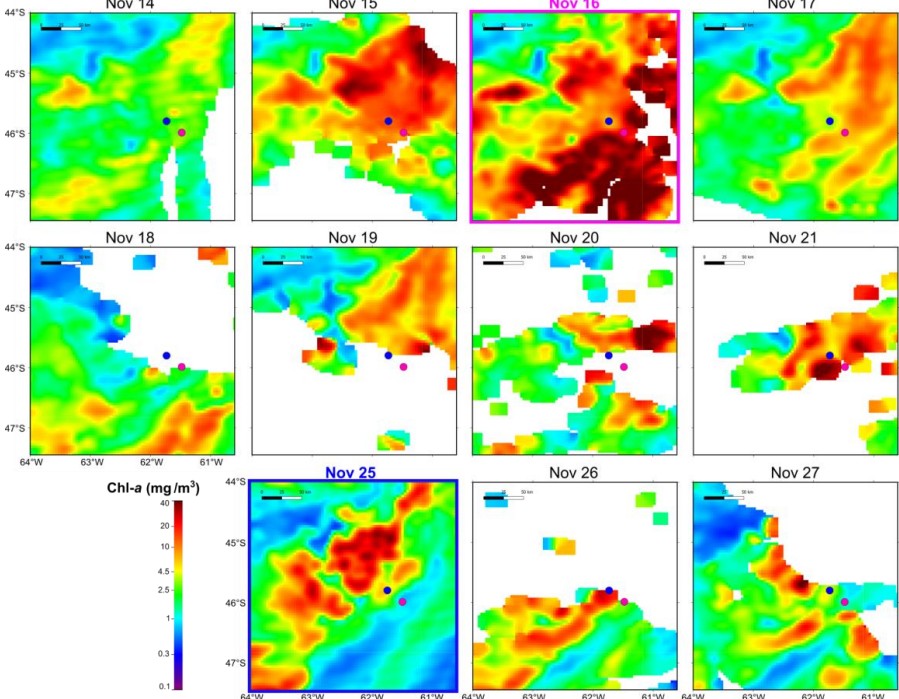

**Figure 3: Daily satellite-derived surface chlorophyll-*a* in the area of the sampling stations: GA01 (pink dot, sampled on November 16, 2021), and AA09 (blue dot, sampled on November 25, 2021).**





**3.2 *In situ* biogeochemical properties and water column structure**

A reddish water discoloration was observed in the bloom area (45.5-46°S, 62-61°W) during the cruises. Surface water temperature and salinity remained similar at both stations GA01 and AA09 sampled with a ten-day interval **(Fig. 4, Table 1)**. Moreover, both stations displayed the same vertical structure in terms of temperature and salinity **(Fig. 4)**, the mixed layer depth (MLD), and the subsurface chlorophyll maximum (SCM) **(Table 1)**. *In situ* chlorophyll-*a* concentration at the surface was 20 µg L$^{-1}$ at GA01 and 4.5 µg L$^{-1}$ at AA09 **(Table 1)**. Dissolved inorganic macronutrients in surface waters were similar in both stages of the bloom, except for nitrate and silicate, which were higher at AA09. In particular, the silicate recovered by ~5 times towards the advanced stage of the bloom.

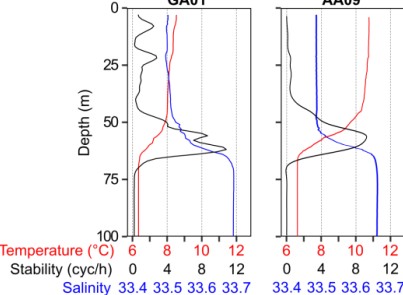

**Figure 4: Vertical profiles of temperature, salinity, and stability measured at the two blooming stations, GA01 (sampled on 16 Nov) and AA09 (sampled on 25 Nov). Strong stratification is denoted by the water column stability (Brunt-Väisäla buoyancy frequency) which indicates a mix layer depth (MLD) around 60 m.**

**Table 1: Surface values of physical and chemical variables measured at stations GA01 and AA09. Sea surface temperature (SST), sea surface salinity (SSS), and concentration of *in situ* chlorophyll-*a* (Chl-*a*), and macronutrients. The depth of the subsurface chlorophyll maxima (SCM) and the mix layer depth (MLD) is also displayed.**

|  | **GA01** | **AA09** |
|---|---|---|
| Date | 16-Nov | 25-Nov |
| SST (°C) | 10.5 | 10.9 |
| SSS | 33.5 | 33.5 |
| SCM (m) | 15 | 10-25 |
| MLD (m) | 60 | 55 |
| Chl-a ( µg L$^{-1}$) | 20.0 | 4.5 |
| Nitrite (µM) | 0.48 | 0.33 |
| Nitrate (µM) | 1.77 | 3.57 |
| Ammonium (µM) | 1.32 | 1.32 |
| Silicate (µM) | 4.54 | 22.35 |
| Phosphate (µM) | 0.33 | 0.44 |

**3.3 Multispecificity of the Amphidomatacean bloom and azaspiracids**

Total protistan plankton of cell size larger than 5µm reached up 31.68 x10$^6$ cells L$^{-1}$ at station GA01, and 10-days later at station AA09, the abundance was 13.69 x10$^6$ cells L$^{-1}$. Of all this total protist abundance, the Amphidomataceae clade represented up to 99 % and 98 %, respectively **(Fig. 5)**, mostly dominated by the non-toxigenic species *Azadinium spinosum* ribotype C and *Azadinium dalianense* **(Fig. 5 and 6)**, representing together >95 % of total Amphidomataceae **(Fig. 5)**. Taxonomic identification up to species level was possible after exhaustive morphological examination of cells under light



microscopy **(Fig. 5)** and scanning electron microscopy **(Fig. 6)**. In the **Appendix A**, more micrographs of Amphidomataceans
are shown taken under light microscopy and scanning electron microscopy **(Figs. A01 to A10)**, along with the rationale for
the Amphidomatacean species designations. The ITS-based metabarcoding of species diversity detected in the field samples
at GA01 and AA09 are also shown in the **Appendix B**, which also support the dominance of *Azadinium spinosum* ribotype C
and *Azadinium dalianense*. In the detailed counting of protistan species in 10-mL subsamples, the well-recognized
Amphidomataceae species under inverted light microscopy for their individual quantification were: *Azadinium spinosum*
ribotype B and ribotype C, *Az. dalianense*, *Az. obesum,* and the smaller taxa *Az. dexteroporum, Amphidoma parvula* and *Am.*
*languida* **(Fig. 5)**. This distinction was based on morphological aspects combining the cell size and shape, such as the
length/wide relation, and other taxonomic aspects. For instance, a slender shape: *Az. spinosum* ribotype B; round: *Az. obesum*,
short, tiny: *Az. dexteroporum, Amphidoma parvula* and *Am. languida*; with a bump in the hypotheca and a pyrenoid in the
hyposome: *Az. dalinaense*; with a spine in the hypotheca: *Azadinium spinosum* ribotype C, *Az. dalianense*, and *Az.*
*dexteroporum*. These and other taxonomic features were further examined by SEM **(Fig. 6)**, for example the number and
arrangement of the thecal plates, the presence of thecal pores, etc. (see the **Appendix A**). Finally, other protists than
Amphidomataceans **(Fig. 7)** contributed up to 1.0 and 2.2 % of the total abundance at stations GA01 and AA09, respectively.
Most of the other protists were heterotrophic and mixotrophic dinoflagellates and ciliates. No diatoms were observed in the
field samples. For an overview of the pure Amphidomataceae bloom in the field samples (e.g. no mucus formation, no
aggregates, cells undergoing cell division), low-magnification micrographs obtained through light microscopy are presented
in the **Appendix C**. The screen of all known and potentially novel variants of azaspiracids, which are produced by
Amphidomataceae, revealed the presence of solely azaspiracid-2 (AZA-2) in both stages of the bloom, with field values of
2122 pg L$^{-1}$ at GA01 and 620 pg L$^{-1}$ at AA09.

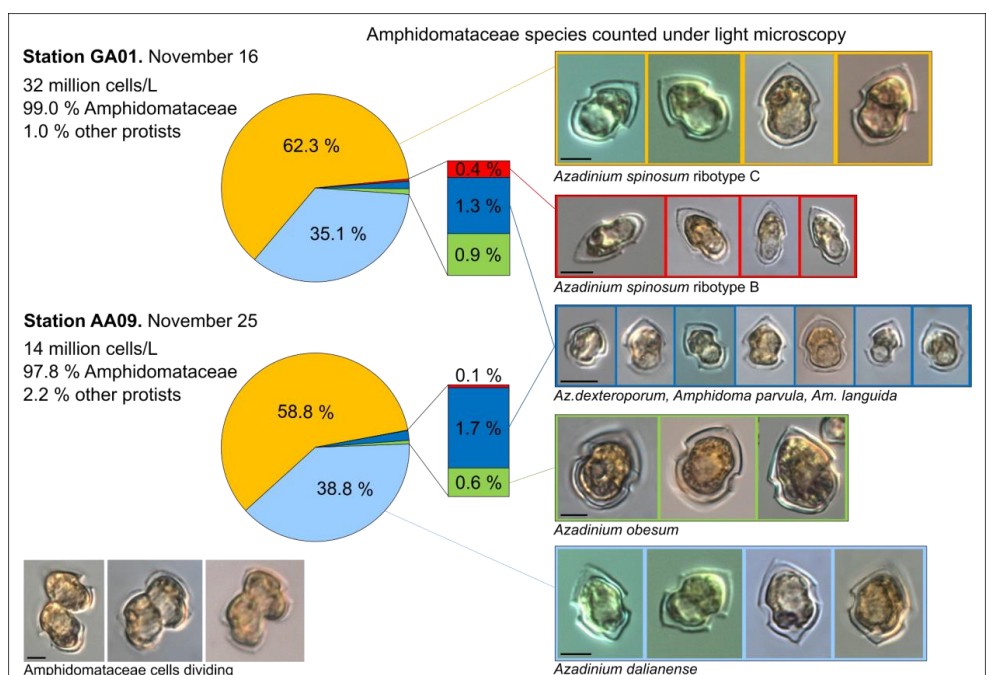

**Figure 5: Relative abundance (in %) of the Amphidomataceae species identified under light microscopy at stations**
**GA01 and AA09. The total abundance of protists at each station was 32 million cells L$^{-1}$ and 14 million cells L$^{-1}$,**
**respectively. From the total cells counted, Amphidomataceae represented up to the 99.0 % and 97.8 %, respectively.**
**The colours in the pie charts correspond to the same species at both stations. Scale bar: 5 μm.**



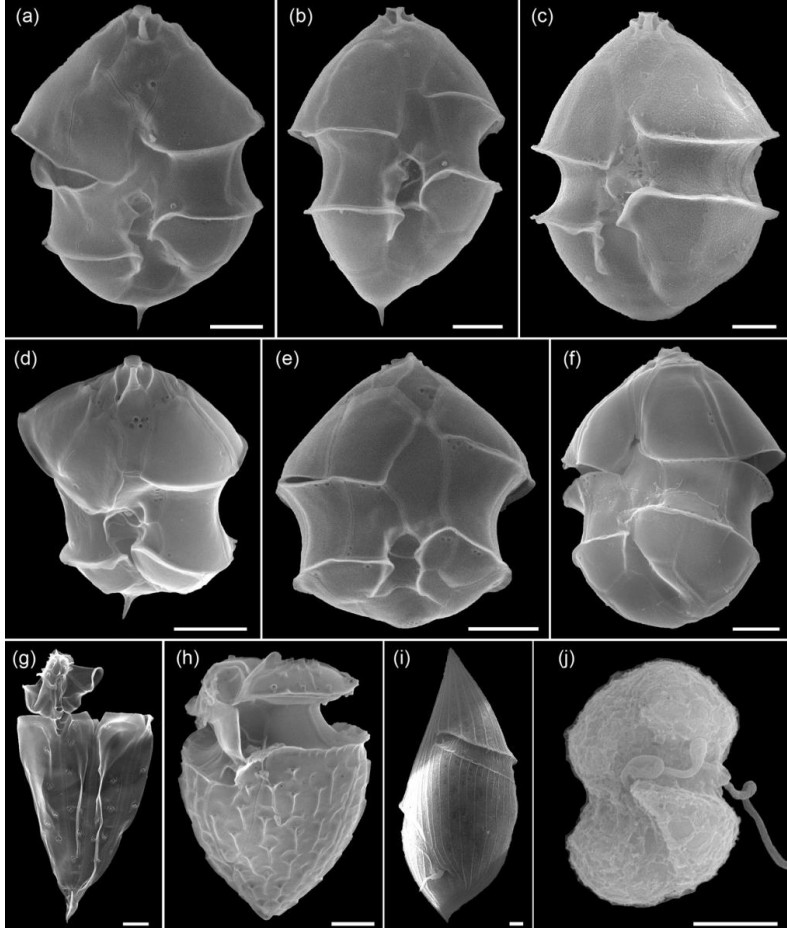


**Figure 6: Scanning electron microscopy of Amphidomataeae species (a-f) and other dinoflagellates (g-j) at stations**
**GA01 and AA09. (a)** *Azadinium spinosum,* **(b)** *Az. dalianense*, **(c)** *Az. obesum*, **(d)** *Az. dexteroporum*, **(e)** *Amphidoma*
*parvula*, **(f)** *Am. languida*, **(g)** *Oxytoxum gracile*, **(h)** *Oxytoxum laticeps*, **(i)** *Gyrodinium* **sp., (j) unidentified gymnodinoid**
**species. Scale bars = 2 μm. See the Appendix A for more micrographs of Amphidomataceans and for evidence and**
**rationale for the Amphidomatacean species designations.**

364

365

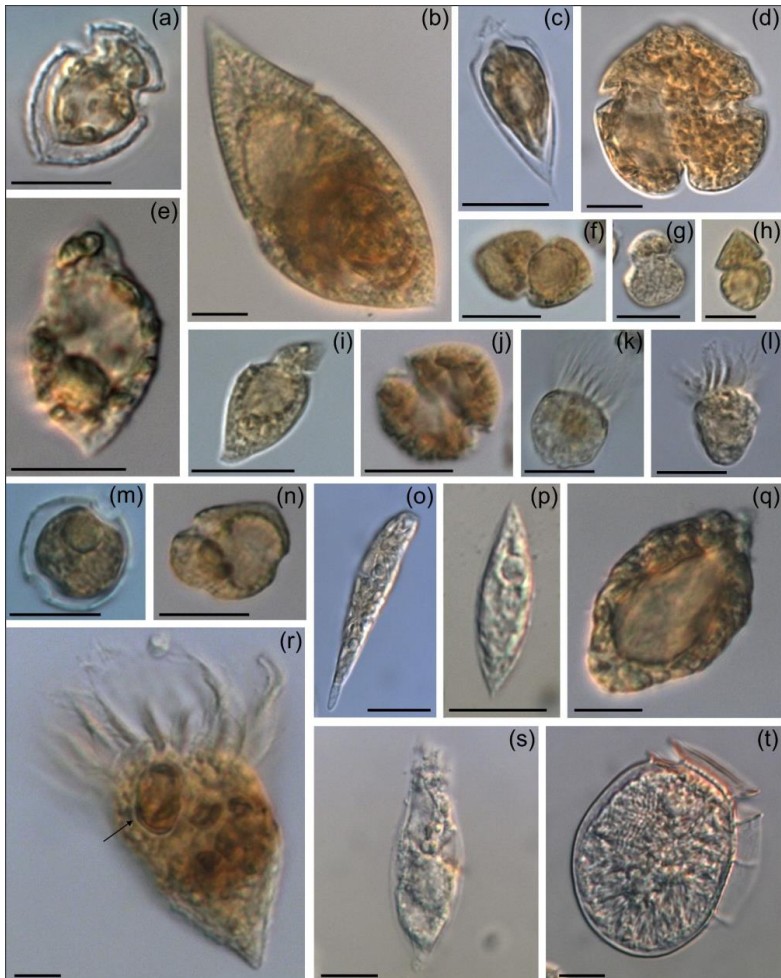

**Figure 7: Micrographs of other protists (1.0 - 2.2 % of total abundance) present in the bloom. a)** *Oxytoxum laticeps*, **b)** *Gyrodinium spiralis*, **c)** *O. gracile*, **d)** *Karenia* **sp., e) unidentified dinoflagellate, f)** *Gyrodinium sp.*, **g-i) unidentfied gymnodinoid cells, k-l) naked ciliates, m)** *Peridinella* **sp., n) Kareniacea-type cell, o) Euglenophyte, p)** *Lessardia elongata*, **q)** *Torodinium robustum*, **r) ciliate with an Amphidomataeae cell (arrow), s) ciliate, t)** *Dinophysis* **sp. Scale bar: 10 μm, except in photos g) and h), scale bar 5 μm.**

**3.4 Surface currents, particle advection model and Lyapunov frontal systems**

The mean and standard deviation of the ADT during the sampling period (16 to 28 November, 2021) evidenced a mesoscale anticyclonic eddy of about 100 km in diameter in the area where the Amphidomataceae bloom was observed at the two sampling locations (**Fig. 8**). In addition, the modelled trajectory of the particles released along the zonal transect at 46°S on November 10 showed high retention at the blooming area over the continental shelf after running for 20 days. On the contrary, high advection within the flow of the Malvinas Current was evidenced (**Fig. 9**). Notably, in the eddy area, particles slightly displaced southwards, remaining trapped in the area after 20 days since their release. The particles advected by geostrophic velocities suggest that the anticyclonic eddy acted as a potential mechanism to retain the Amphidomataceae bloom within the same location during the two synoptic samplings (**Fig. 9**). All the other particles released East and West of the eddy displayed





a different behaviour (**Fig. 9**), also shown in the **Appendix D** where four parcels of particles were advected from 16 to 25
November. West of the eddy they were advected northward at rather slow speed (average 5 cm/s) while East of the eddy they
increased their speed towards the north as they approached the continental slope. Within the core of the Malvinas Current,
speeds were as large as 80 cm/s (**Fig. 9**).

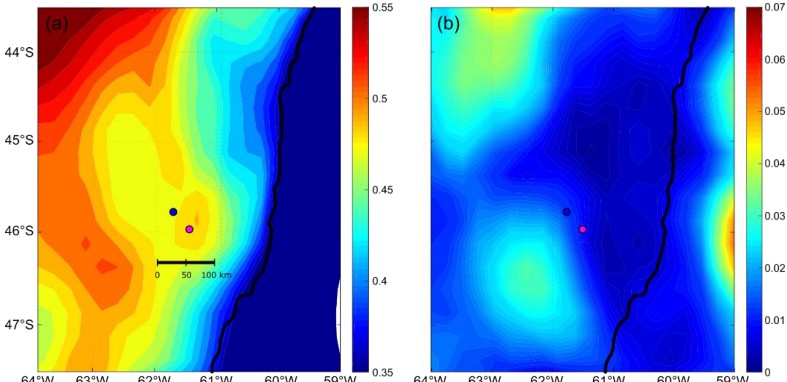

**Figure 8: (a) Mean and (b) standard deviation (std) of the Absolute Dynamic Topography (ADT, in meters) displayed**
**in colour scale, during the period November 16 to November 28, embracing the sampling period on both bloom stations:**
**GA01 (pink dot, Gayoso cruise) and AA09 (blue dot, Agujero Azul cruise).**

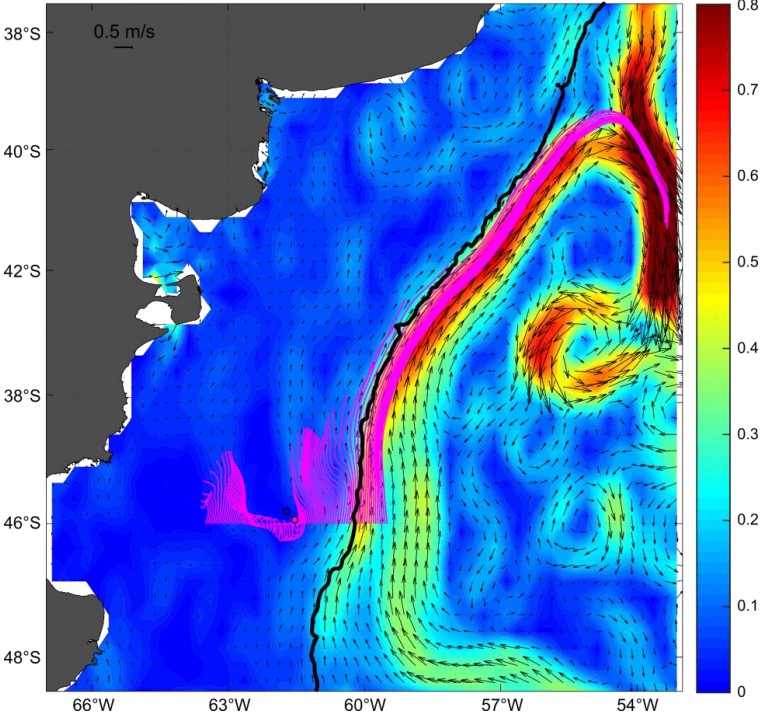

**Figure 9: Particle advection after 20 days since particle release on November 10, 2021. Initial points: one particle every**
**0.05 grades along 46°S. Note the high retention and the distinct behaviour of particles in the area of the sampling**
**stations: GA01 (pink dot) and AA09 (blue dot). Satellite altimetry geostrophic velocities averaged for the same 20 days**
**are also displayed with black vectors; the magnitude of the averaged speeds is represented with colours in the**
**background (m/s).**



Moreover, the finite-size Lyapunov exponent (FSLE) ridges highlighted the stirring and hydrologically complex nature of the
Southwestern Atlantic Ocean, associated with the high hydrographical heterogeneity of the oceanic waters (**Fig. 10**). Although
FSLE were less intense over the shelf than in the adjacent oceanic waters, in the area where the Amphidomataceae bloom was
sampled (pink dashed square in **Fig. 10**), two relatively strong FSLE ridges consistently kept both bloom stations within the
same water mass during the period between the two synoptic samplings (**Fig. 11**).

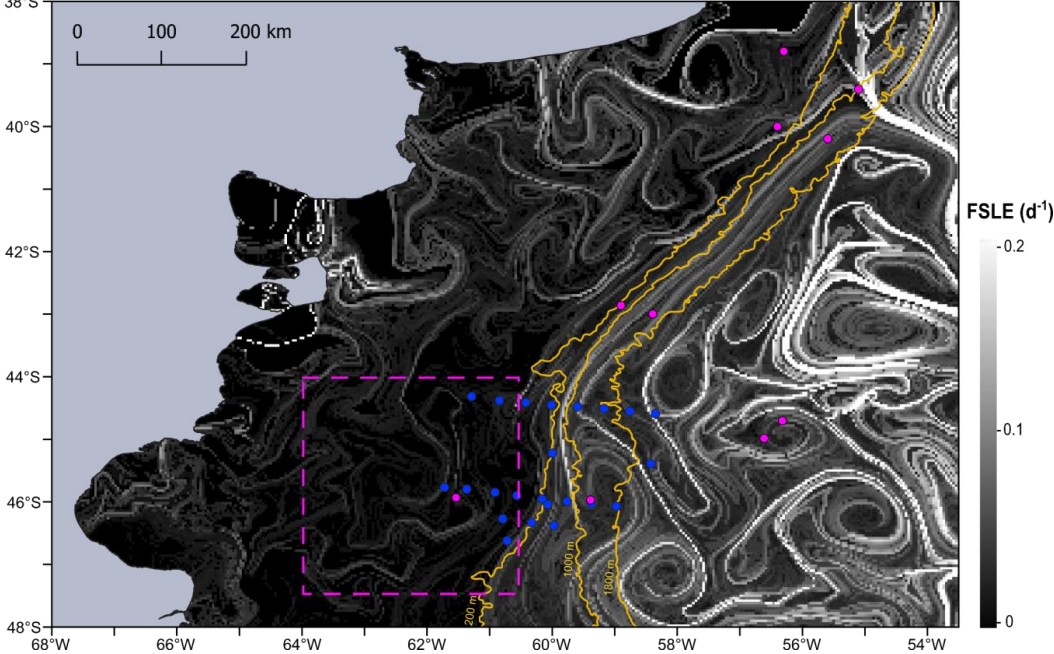


**Figure 10: Fronts identified as Finite-size Lyapunov Exponent (FSLE) ridges computed for November 16, shown**
**against a grayscale background. All sampling stations from the Gayoso cruise (pink dots) and Agujero Azul cruise**
**(blue dots) are indicated. November 16 corresponds to the sampling of the Amphidomataceae bloom at station GA01**
**(pink dot within the square marked by a dashed pink line). The square highlights the area shown in Fig. 11.**



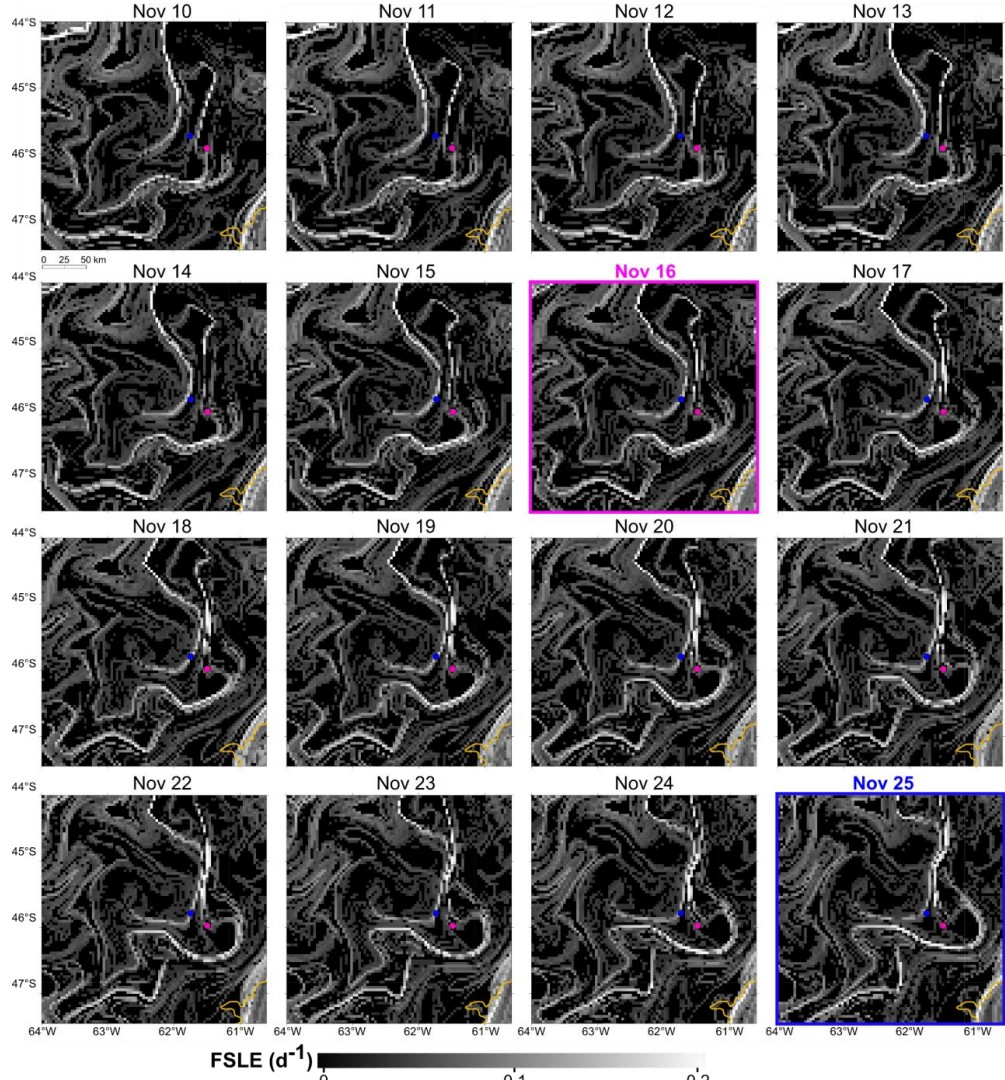

**Figure 11: Finite-size Lyapunov Exponent (FSLE) ridges in the area of the two locations with the Amphidomataceae bloom: GA01 (pink dot) sampled on November 16, and AA09 (blue dot) sampled on November 25. See the Video of these daily images in the Appendix D. The 200 m isobath is indicated in yellow.**

### 4. Discussion

#### 4.1. Amphidomataceae blooms in the Patagonian shelf

The phytoplankton spring and summer blooms in the Patagonian shelf display a southward progression related to the seasonal thermal cycle. In early spring (September-October), the water column stratifies north of ~45°S, favouring the proliferation of opportunistic micro-diatoms in the nutrient-rich, well-lit surface layers (Ferronato et al., 2023). South of ~45°S, the bloom initiates later in spring-early summer (December-January) and continues until autumn (March) (revised in Guinder et al., 2024). Here, blooms of nanoflagellates and dinoflagellates are triggered by combined vertical stability and nutrient-depleted surface waters (especially silicates) after the early-spring blooms of large diatoms (Balch et al., 2014; Carreto et al., 2016;



Ferronato et al., 2024). The massive proliferation of Amphidomataceans in mid-November 2021 was in line with this
successional pattern. These nano-dinoflagellates bloomed in a stratified water mass with a deep mixed layer depth (~60 m).
While dinoflagellates are less effective at nutrient resorption compared to diatoms, they can move throughout the stable water
column to find light and nutrients (Glibert, 2016), especially at low phosphate levels (Lin et al., 2016) as observed at the
stations GA01 and AA09 with high nitrate-to-phosphate ratios.
The success of the multispecific bloom of Amphidomataceae may be attributed to a combination of multiple intrinsic and
extrinsic factors. For instance, their small cell-size, unique swimming modes and the production of azaspiracids may have
alleviated grazing pressure (Tillmann et al., 2019). In fact, in the fixed samples, it was observed that many cells were obviously
active and undergoing cell division and that these nanoflagellates were overwhelmingly the predominant photosynthetic
protists responsible for the high chl-$a$ levels, with negligible abundance of micro-grazers accompanying the bloom
development. The presence of less than 2% of other protists (mixotrophs and heterotrophs) could be also related to a delay in
the recovery of predators following the early blooms of microdiatoms, as well as to an abrupt development of the
Amphidomataceae bloom, which may have prevented micrograzers from taking advantage of the available food. Another
observation supporting the active persistence of the bloom was the pristine condition of the microenvironment surrounding the
dense populations of Amphidomataceae (see **Appendix C**), with no aggregates or mucus formation typically observed in the
late stages of blooms (Genitsaris et al., 2021). Furthermore, no competitors for light and nutrients were detected in the samples;
specifically, no diatoms were found during microscopic examination. This may explain the rapid recovery of silicates (from 5
to 22 µM) over the 10-day persistence of the Amphidomataceae bloom, as these silicates were not being utilised by silicate-
requiring species. A similar observation was noticed during a bloom induced by iron-fertilization in the Southern Ocean, where
diatoms predominated and silicate levels decreased from 10 to 6 µM within the bloom patch over the course of 12-days
(Abraham et al., 2000).

**4.2. Highest abundance ever recorded for a bloom of Amphidomataceans**

With 17 described species of *Azadinium* and 14 of *Amphidoma*, the Amphidomataceae represent a small but diverse group of
dinoflagellates. Most of these species are very similar in size and shape, which makes the qualitative identification and, in
particular, the species specific quantification in field samples difficult. Hence, characterising the cryptic species of the
multispecific bloom of Amphidomataceae in this study represented a major challenge, where a reliable species identification
was achieved by the combination of several diagnostic details using electron microscopy. Light microscopic counting of fixed
samples provided high quantitative accuracy, but reliable species identification was only possible in a few cases.
Complementing the identification by microscopy with metabarcoding specifically targeting the internal transcribed spacer
(ITS1) region (Liu et al., 2023), allowed for the detailed characterization of Amphidomatacean species diversity in the field
samples. By combining the three approaches we were able to identify ribotypes and the toxic species, including previously
described Amphidomataceae species for the Argentine Sea (Fabro et al. 2019; Tillmann et al., 2019; 2021) or species still
undescribed in the global seas (see the **Appendix A**).
In the North Atlantic, AZA-1 (and its producing species, *Az. spinosum* ribotype A), has been identified as one of the most
prevalent toxins among a wide range of AZA variants (Tillmann et al. 2021). Bloom density of Amphidomataceae around
Ireland have been reported as 8.3 x $10^4$ cells L-1 for *Az. spinosum* (Wietkamp et al. 2020) and 47 x $10^5$ cells L$^{-1}$ for *Amphidoma*
*languida* (McGirr et al. 2022) and a small bloom of *Am. languida* in the North Sea with 1.2 $10^5$ cells L$^{-1}$ has also been described
(Wietkamp et al. 2020). In this region, cases of human intoxication with azaspiracids have been linked to the consumption of
contaminated mussels from the Irish West coast, where blooms in the shelf-break area can reach coastal shellfish beds through
wind-driven advection (Raine, 2014). Notably, in the Argentine Sea, only AZA-2 has been detected in field and culture samples



(Turner and Goya, 2015; Fabro et al., 2019; Tillmann et al., 2019; Guinder et al., 2020), and so far, no poisoning events have
been attributed to AZAs.
In this study, relatively high levels of solely AZA-2 were detected in bloom samples. A toxin profile of solely AZA-2 is up to
now only known for the Argentine strain H-1-D11 of *Azadinium spinosum* ribotype B (Tillmann et al., 2019), and this ribotype
was also identified in the present bloom. Relating AZA quantities to the relatively low abundance (0.1 to 0.4 % of total
Amphidomataceae) of *Az. spinosum* ribtype B revealed AZA-2 cell quotas of 17-42 fg per cell, which is an order of magnitude
higher that the cells quota of 2 to 9 fg per cell for strain H-1-D11 grown under laboratory conditions (Tillmann et al. 2019). In
fact, *Az. spinosum* of ribotype A producing AZA-1 and -2 is the primary causative agent of AZA poisoning in Europe
(Tillmann, 2018). However, the large majority of cells of *Az. spinosum* in the present bloom sample is from ribotype C, which
is, based on analyses of several strains from Argentina, non-toxigenic (Tillmann et al 2019), and all globally available strains
(including strains from Argentina) of the other co-dominant species in the bloom, *Az. dalianene*, (Tillmann et al., 2019), also
do not produce azaspiraids. In the Chilean continental shelf in the SE Pacific, AZAs have been detected in scallops and mussels
(López-Rivera et al., 2010), but no intoxication events or large blooms of this clade have been documented. Moreover, only
*Az. poporum* has been described as an AZA producer in Chilean waters (Tillmann et al., 2017a). Likewise in Peru, a relatively
high bloom (up to one million cells per liter) of *Az. polongum* was detected in the summer of 2014, with no AZA production
(Tillmann et al., 2017b). Although the continental shelves of Chile-Perú and Argentina have different hydrology, both span
similar latitudinal gradients along the South American coasts and are influenced by the Humboldt and Malvinas Currents,
which share a common origin in the Circumpolar Antarctic Current. Strikingly, both shelves exhibit different populations of
Amphidomataceae, despite the expectation that ocean currents could serve as transport pathways for HAB species, promoting
their dispersion (Giddings et al., 2014).

**4.3. Spatio-temporal evolution of the bloom: retention and stirring**

Phytoplankton bloom initiation, magnitude, and persistence rely on a host of biogeochemical and physical processes. As
discussed in previous sections, the explosive onset of the bloom of multiple species of Amphidomataceae was associated with
a combination of water column stability, the negligible presence of micrograzers, and the ecological traits of this group that
facilitate massive proliferation. However, these conditions alone do not fully explain the persistence of this bloom, which was
sampled 10 days later at a location 40 km away in the offshore waters of the Patagonian shelf, where strong surface currents
were expected to disperse plankton blooms. The persistence of this extraordinary bloom, characterized by its remarkable
magnitude and consistent species composition, indicates the retention and accumulation of the bloom patch within the same
water mass. In addition to the biological evidence confirming the presence of the same Amphidomataceae bloom at both
sampling stages, analyses of circulation through altimetry, particle experiments, and FSLE—an indicator of frontal activity
and stirring intensity—support the conclusion that the same bloom patch was captured at both locations. Two potential
scenarios could explain this: (1) the same patch remained in the area over the ten days of sampling, occupying a space of 40
km or larger (**Fig. 12a and b**), or (2) a smaller bloom patch was initially detected at station GA01 and then transported by
stirring towards AA09 (**Fig. 12c**). A less likely scenario is that (3) two Amphidomataceae blooms developed independently at
both locations (**Fig. 12d**). This situation is improbable due to the complex physical-biological interactions that drive different
bloom developments, such as variable stirring (Abraham et al., 2000; Lehan et al., 2007; Della Penna et al., 2015) and changing
environmental conditions that select for different species and functional groups across diverse spatio-temporal scales ( Levy
et al., 2018; Hernández-Carrasco et al., 2020; Mangolte et al., 2023). Moreover, no dormant cysts of Amphidomataceae are
known, which could explain population outbreaks in specific locations, as observed in frontal areas for other dinoflagellates
forming HABs (Smayda, 2002; Akselman et al., 2015).





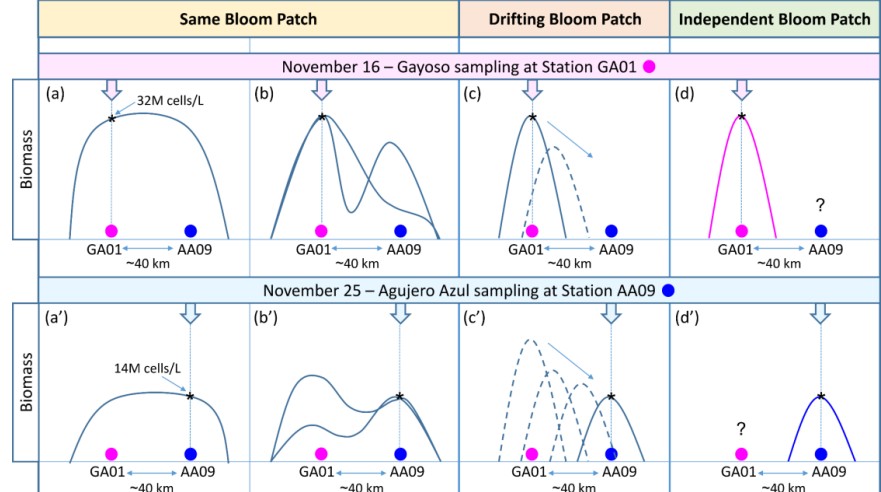

**Figure 12: Hypothetical scenarios of the spatio-temporal evolution of the Amphidomataceae bloom during the 10-day period between the synoptic sampling at the two stations 40 km apart: GA01 (pink circle) and AA09 (blue circle). The bloom's biomass was 32 million cells $L^{-1}$ at GA01 and 14 million cells $L^{-1}$ at AA09. In the Same Bloom Patch (a-a' and b-b'), the bloom covered an area encompassing both sampling stations. The bloom developed such that the biomass was either distributed homogeneously across the patch (a-a') or heterogeneously (b-b'), resulting in variable biomass patterns over time and space. In the Drifting Bloom Patch (c-c'), the water mass with the Amphidomataceae bloom detected at station GA01 was transported by currents towards station AA09, where the bloom was detected with lower biomass but still with high intensity. The Independent Bloom Patch (d-d') suggests that two discrete, autonomous Amphidomatacean blooms developed locally at each sampling station.**

Additionally, the hydrographically complex Southwestern Atlantic creates a variety of microhabitats at the meso- and submesoscale. These include areas of upwelling, downwelling, eddies, retention, and dispersion (Becker et al., 2023; Beron-Vera et al., 2020; Salyuk et al., 2022; Saraceno et al., 2024). This spatial heterogeneity enhances the development of variable nutrient patches and phytoplankton productivity (Lehahan et al., 2017; Levy et al., 2018; Hernández-Carrazco et al., 2020; Ser-Giacomi et al., 2023). During the Gayoso cruise, contrasting phytoplankton assemblages and bloom types were observed at all sampling locations, including distinct blooms of dinoflagellates, coccolithophores, diatoms, and nanoflagellates. These variations were related to substantial heterogeneity in surface velocities and environmental conditions across the region (Ferronato et al., 2024). In this study, while the retention of the Amphidomataceae bloom is certainly limited by the accuracy of satellite altimetry maps, the documented 100 km diameter of the eddy is a reasonable size that can be distinguished using the gridded satellite altimetry maps produced by CMEMS. Although this retention is transient, this particular circulation facilitated the massive development of the Amphidomataceae bloom, with no evidence that this patch was advected through the Malvinas Current, as observed with drifters released east of the bloom area in spring 2021 (Saraceno et al., 2024). Our results highlight the importance of studying the evolution of phytoplankton blooms on continental shelves, focusing on the bio-physical coupling that drives their patchy nature, persistence, and transport, in order to capture short-lived blooms and their potential to cause toxic outbreaks.

Overall, this study is unique from both biological and physical perspectives due to the following factors: (i) the Amphidomataceae bloom observed in spring 2021 in the Argentine Sea, with up to 32 million cells per litre, represents the largest bloom of this clade ever recorded globally; (ii) unusual sampling in offshore shelf waters with two vessels over a ten-day interval allowed for synoptic observations of the bloom at two active developmental stages; (iii) simultaneous ecological





characterization of the bloom and surface currents and fronts provided insights into patch stirring and the short-term evolution
of the bloom; (iv) field quantitative abundance data for Amphidomataceae are rare, and to our knowledge, this is the first
detailed description of species abundance in field samples, combining light microscopy, electron microscopy, and
metabarcoding; (v) the relatively low abundance of *Azadinium spinosum* Ribotype B indicated high AZA-2 cell quotas; (vi)
the fine-taxon assessment of the Amphidomataceae bloom revealed biogeographical patterns and strain-specific toxic
potential; and (vii) the use of interdisciplinary approaches sheds light on the bio-physical coupling underlying the persistence
and horizontal transport of this extraordinary bloom in offshore shelf waters.

**Appendix A**

**A1. Estimation and categorization of Amphidomatacean species diversity**
For the present study, we combined light microscopy (LM) quantification, scanning electron microscopy (SEM) examination,
and metabarcoding to characterize the field samples as accurately as possible, both qualitatively and quantitatively. Generally,
the following species of Amphidomataceae were identified:

**A1.1. *Azadinium spinosum***
Specimens were identified with SEM as *Az. spinosum* based on the combination of (1) presence of an antapical spine, and (2)
presence of a ventral pore located on the right side of the suture of Plate 1' and 1" (Fig. A01). The vast majority of cells thus
identified as *Az. spinosum* had a somewhat broader cell shape. Generally, identification of *Az. spinosum* is complicated as there
are several different ribotypes (Tillmann et al., 2021), which notably differ in azaspiracid toxin presence and profile. In a
previous study from the Argentine shelf region, it was shown that 23 out of 24 isolated strains of *Az. spinosum* were assigned
to the non-toxigenic ribotype group C, and cells from these strains also had a somewhat broader cell shape (Tillmann et al.,
2019). Metabarcoding of the present bloom sample revealed that the most common sequences showed a high match with
sequences of these Argentine ribotype C strains. A dominance of non-AZA-producing *Az spinosum* (ribotype C) also aligns
with the finding that no AZA-1, the marker toxin of the toxigenic *Azadinium spinosum* ribotype A strains (Tillmann et al.,
2021), was detected in the field samples. In the quantitative light microscopy (LM) counts, all medium-sized cells (length >
12 µm) with a rounded hypotheca and an antapical spine were thus categorized as *Azadinium spinosum* ribotype C.

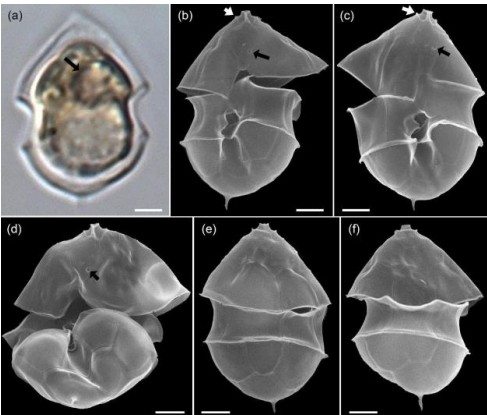
**Figure A01: LM (a) and SEM (b-f) of cells of the Amphidomatacean bloom stations identified as *Azadinium spinosum***
**ribotype C. (b–d) Cells in ventral view. (e, f) Cells in dorsal view. Note the pryrenoid (black arrow in a), the position of**
**the ventral pore (black arrows in b, c, d), the rim around the pore plate (white arrows in b, c) and the distinct antapical**
**spine. Scale bars = 2 µm.**





Recent studies have revealed the presence of a new, molecularly distinct species of *Azadinium* in the North Atlantic, which is
morphologically indistinguishable from *Az spinosum* and is currently provisionally referred to as *Azadinium* cf. *spinosum*
(Tillmann et al., 2021). Therefore, it cannot be ruled out that this species was also present in the samples, but metabarcoding
showed no evidence of the presence of *Az.* cf. *spinosum* in the bloom samples.
In addition to these broader cells of *Az. spinosum* ribotye C cells, LM analysis revealed a (much rarer) presence of distinctly
slender cells with an antapical spine (Fig. A02a). Such cells perfectly correspond to cell shape of a single strain H-1-D11 from
Argentina identified as a ribotype B strain of *Az spinosum* (Tillmann et al., 2019), and this strain is depicted here for comparison
(Fig. A02c, d). In SEM, specimen of slender shape lacked the rim around the pore plate (Fig. A02b), which is the morphological
diagnostic feature differing in ribotype B strains from ribotype A and C strains, which all have a thick rim. Additionally,
metabarcoding showed conformity of some sequences with other ribotype B *Az. spinsum* strains (e.g. 99% similarity with the
Argentinean strain H-1-D11). Consequently, all slender cells with an antapical spine quantified in LM were categorized here
as *Az. spinosum* ribotype B. Strain H-1-D11 from Argentina was shown to produce solely AZA-2 (Tillmann et al., 2019). As
this was the only AZA detected in our field sample, this is additional support for this *Az. spinosum* ribotype B designation.

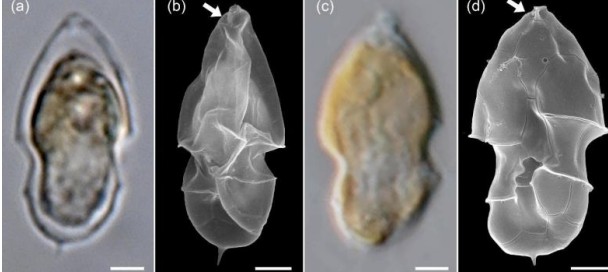


**Figure A02: LM (a), and SEM (b) of cells of the amphidomatacean bloom stations identified as *Azadinium spinosum*.**
**For comparison, LM (c) and SEM (d) of cells of strain H1-D11 of *Azadinum spinosum* ribotype B isolated from the**
**Argentine shelf in 2015. Note the elongated cells shape, the distinct antapical spine, and the lack of a rim around the**
**pore plate (white arrows in b and d). Scale bars = 2 μm.**

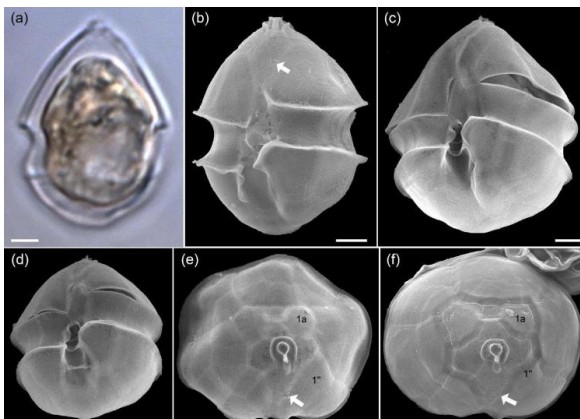


**Figure A03: LM (a) and SEM (b-f) of cells of the Amphidomatacean bloom stations identified as *Azadinium obesum*.**
**(b–d) Cells in ventral view. (e, f) Epitheca in apical view. Note the lack on an antapical spine, the position of the ventral**
**pore (white arrows in b, e, f) and the lack of contact between Plates 1a and 1'' (kofoidian plate label notation) visible in**
**e) and f). Scale bars = 2 μm.**




**A 1.2. *Azadinium obesum***

Cells of the non-toxigeneic species *Az. obesum* were identified in the SEM samples based on the combination of the following features: (1) no antapical spine, (2) ventral pore on the right side of Plate 1', and (3) no contact between Plates 1a and 1" (Fig. A03). All such cells had a distinctly broad oval shape and were relatively large. In the light microscope, all relatively large oval Amphidomataceae cells with a rounded hypotheca and no visible spine were therefore categorized as *Azadinium obesum*.

617

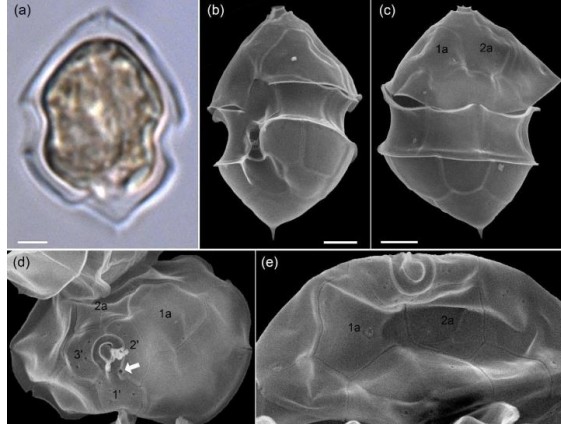

618

**Figure A04: LM (a) and SEM (b-f) of cells of the amphidomatacean bloom stations identified as *Azadinium dalianense*. (b) Cell in ventral view. (c) Cell in dorsal view. (d) Epitheca in apical view. (f) Epitheca in dorsal view. Note the distinct apical spine on a triangular bumpy hypotheca (a–c), the position of the ventral pore on the left side of the pore plate (white arrow in d), the presence of only two large anterior intercalary plates 1a and 2a (c–e) and presence of only 3 apical plates in (d) (kofoidian plate label notation). Scale bars = 2 μm.**

624

**A 1.3. *Azadinium dalianense***

With the rounded hypotheca, cells of *Az. obe*sum and *Az. spinosum* ribotype C were clearly distinguishable in the light microscope from cells with a distinctly protruding bump, at the tip of which a small spine was present (Fig. A04a). Corresponding cells detected in SEM preparations (Fig A04b–e) were identified as *Az. dalianense*, based on the combination of the following features: (1) ventral pore on the left side of the pore plate, (2) asymmetrical hypotheca with a bump and a distinct antapical spine, and (3) presence of only 3 apical plates and 2 anterior intercalary plates. Since all three features were only rarely visible simultaneously due to the cell's orientation, the presence of the somewhat similar species *Az. perfusorium* cannot be ruled out. *Az. perfusorium* also has a posterior small bump with a spine and a ventral pore located on the left side of the pore plate, but it possesses 4 apical plates and 3 intercalary plates (Salas et al., 2021). However, neither the SEM nor metabarcoding provided any indication of its presence in the samples. The occurrence of *Az. dalianense* in the region is well documented by a series of strains isolated from the Argentine shelf in 2015 (Tillmann et al., 2019), and *Az. dalianense* was also identified as part of *Azadinium* blooms in 1991(Tillmann and Akselman, 2016). Metabracoding additionally indicated the presence of two differen ribotypes of *Az. dalianense*, namely E and B as defined in Tillmann et al. (2019), where all previous strains from Argentina belong to the ribotype E clade. Accordingly, the majority of reads from the bloom station *Az. dalianense* were from ribotype E (represented by strains H-4-E8 and N-12-04 in the reference dataset), whereas reads of ribotybe B (represented by strain IFR-ADA-01C) made only ca. 0.02 % of all *Az. dalianense* reads. All strains of *Az. dalianese* representing different ribotypes collected from various regions analysed so far were non-toxigenic.

642



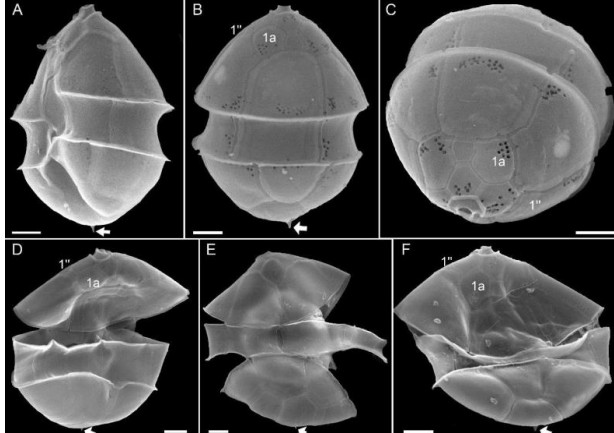

643

**Figure A05: SEM of yet unidentified cells of *Azadinium* sp. 1 of the Amphidomatacean bloom stations. (a) Cell in left-lateral ventral view. (b) Cell in dorsal view. (c) Cell in apical view. (d–f) Cells in dorsal view. Note the small indistinct apical spine (white arrow in a, b, d–f) and the lack of contact between plates 1a and 1'' (kofoidian plate label notation) visible in b, c, d, f. Scale bars = 2 μm.**

The classification and quantification of *Az. obesum* and *Az. dalianense*, however, is complicated by the fact that a number of cells of an unclear assignment were found in the samples (Fig. A05). These cells, like *Az. obesum*, (1) had a ventral pore on the right side of Plate 1' and (2) no contact between Plates 1a and 1'', but unlike *Az. obesum*, they had a distinct, albeit small, antapical spine. This combination of features is not known from any described *Azadinium* species, suggesting that this may be a new species. However, for a complete description as a new *Azadinium* species, further investigations are necessary, ideally adding sequence data and analyses of toxin production. In any case, it is clear that cells of this type will have been included in the categories *Az. obesum* or *Az. spinosum* during the light microscope analyses and quantifications.

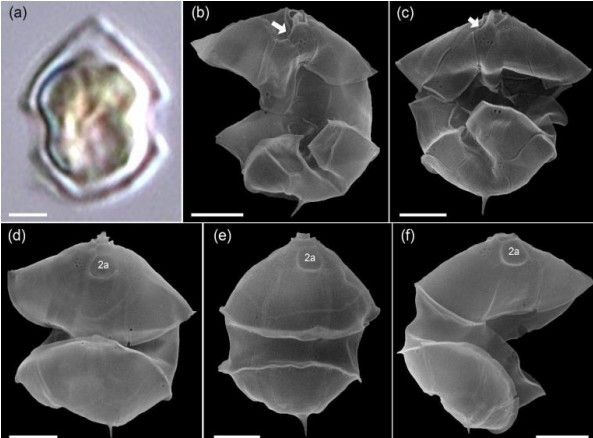

656

**Figure A06: LM (a) and SEM (b-f) of cells of the Amphidomatacean bloom stations identified as *Azadinium dexteroporum*. (b, c) Cells in ventral view. (d–f) Cells in dorsal view. Note the small size, the distinct antapical spine, the position of the ventral pore on the right side of the pore plate (white arrows in b, c,), and the concave central intercalary plate 2a visible in d–f. Scale bars = 2 μm.**

**A 1.4. Smaller Amphidomatacean species: *Azadinium dexteroporum*, *Amphidoma parvula*, and *Amphidoma languida***

While *Az. spinosum*, *Az. obesum*, and *Az. dalianense* fall into a slightly larger size class, a number of smaller *Azadinium* species





were identified and categorized in the samples. One of them was *Az. dexteroporum* (Fig. A06), which was identified in the
SEM by the following combination of features: (1) relatively small size, (2) presence of a distinct antapical spine, (3) a slightly
posteriorly positioned ventral pore on the right side of the ventral plate, and (4) a distinctly concave central intercalary Plate
2a. Metabarcoding revealed a number reads for an *Azadinium* sp. 1 with *Az. dexteroporum* as closest species suggestion (line
7 in Table S01), however only with rather low similarity (90-95%) compared to the reference database. Global wise, there are
only three available strains of *Az. dexteroporum*, and of those only one strain from the Mediterranean was identified as a
producer of AZA (Rossi et al. 2017). In contrast, two additional strains from the North Atlantic, which also had marked
sequence differences compared to the Mediterranean strain, did not produce AZAs (Tillmann et al. 2020). The low similarity
of *Az. dexteroprum* reads from the present bloom samples thus indicate that the local population may represent a new ribotype
quite distinct from the AZA-producing ribotype, and strain isolation of local *Az. dexteroporum* is needed to clarify its identity
and toxin production potential.

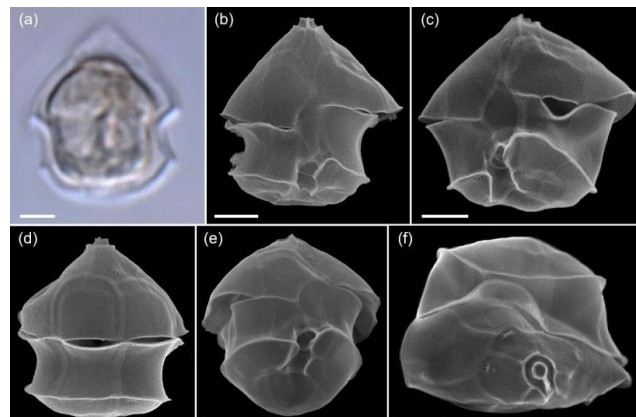


**Figure A07: LM (a) and SEM (b-f) of cells of the Amphidomatacean bloom stations identified as *Amphidoma parvula*.**
**(b, c) Cells in ventral view. (d) Cell in dorsal view. (e) Cell in ventral antapical view. (f) cell in apical view. Note the**
**small size, the flat hypotheca, the shape of the 1' plate visible in a, b, the group of pores in the second antapical plate**
**(white arrow in e), and the relatively long apical plates visible in d and f. Scale bars = 2 μm.**

In the same size class, cells were also observed that were identified in SEM as *Amphidoma parvula* (Fig. A07) by (1) their flat
hypotheca and (2) a characteristically shaped 1' plate. This non-toxigenic species was described in 2018 based on a culture
isolated from the Argentine shelf (Tillmann et al., 2018). In accordance, a low number of reads with high similarity to *Am.*
*parvula* strain H-1E9 (>98 %) recorded by metabarcoding. With its relatively long apical plates, *Am. parvula* could also be
easily distinguished in the SEM from the similarly small *Am. languida*, which was also identified in SEM (Fig. A08). In *Am.*
*languida*, (1) the small apical plates, and (2) the presence of a large characteristic antapical pore is a distinguishing feature.





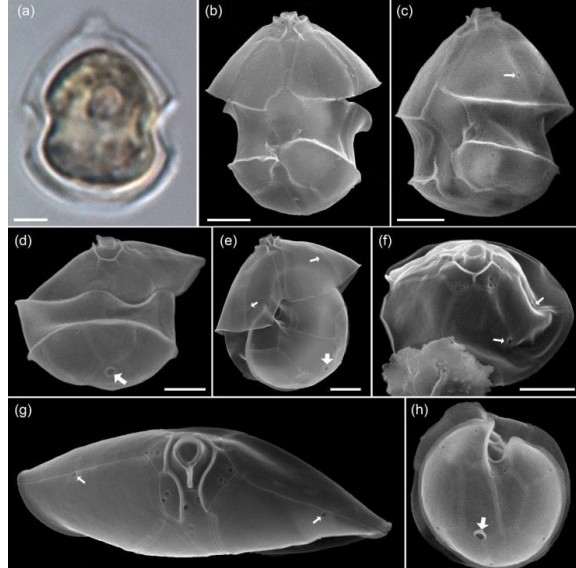

**Figure A08: LM (a) and SEM (b-f) of cells of the Amphidomatacean bloom stations identified as *Amphidoma languida*.**
**(b, c) Cells in ventral view. (d) Cell in dorsal view. (e) Cell with epitheca in ventral view and hypotheca in antapical**
**view. (f) Epitheca in lateral apical view. (g) epitheca in apical view. (h) Hypotheca in antapical view. Note the shape of**
**the 1' Plate visible in a), the distinct antapical pore in the second antapical plate (white arrows in d, e, h), and the**
**relatively short antapical plates visible in d, f, g. Also note that there are only single pores on precingular plates (small**
**white arrows in c, e, f, g). Scale bars = 2 μm.**

However, in 2024, a new species of *Amphidoma*, *Am. fulgens*, was described, which is morphologically almost identical to *Am. languida* but shows significantly different sequence data and, different to *Am. languida*, does not produce azaspiracids (Kuwata et al., 2024). *Amphidoma fulgens* was found to be widely distributed in the Pacific, but there are no records yet from the Atlantic Ocean. Despite its presence in the bloom samples, there were no reads related with higher similarities to *Am. languida* in the ITS metabarcoding data set, but only very few ready with rather low similarity to *Amphidoma entries in the database-* This seem to be in line with the general notion that ITS sequencing of cultured strains of *Am. languida* in many cases failed (Wietkamp et al. 2019). In fact, data of a previous metabarcoding study (Liu et al. 2023) showed that *Am. languida* hits were abundant in Chinese waters in the LSU dataset but absent in the ITS-1 data set. While *Am. languida* and also *Am. fulgens*, according to their original descriptions, only have one small pore on each precingulate plate (Tillmann et al., 2012; Kuwata et al., 2024), there were also several cells of *Amphidoma* with three or more small pores on individual precingular plates (Fig. A09a–d). To what extent these cells represent *Am. languida* or other yet undescribed and closely related species, requires further clarification.



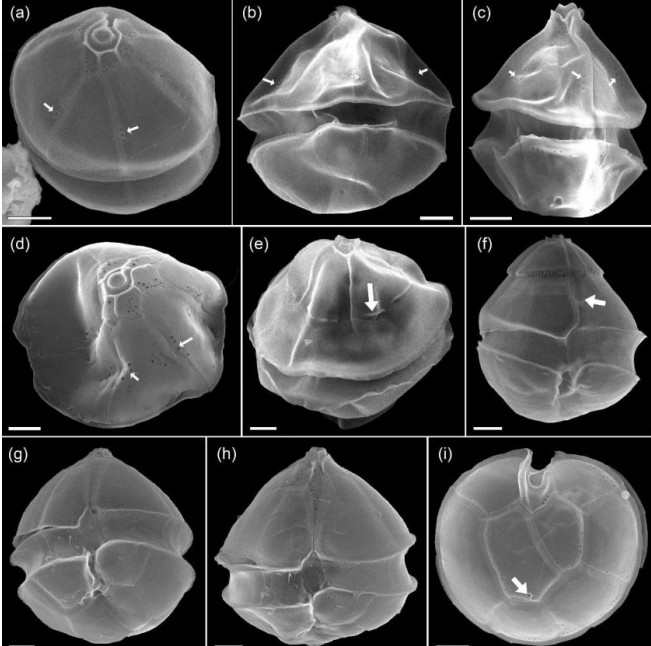


**Figure A09: SEM of unidentified cells of *Amphidoma* spp. of the Amphidomatacean bloom stations. (a–d) Cells in apical**
**(a, c) or dorsal (b, c) view resembling *Amphidoma languida*, but with multiple pores in precingular plates (small white**
**arrows in a–d). (e) A cell of *Amphidoma* sp. in dorsal view. Note the very long apical plates. The row of pores with a**
**distinct rim on the apical plates (white arrow in e) resemble *Amphidoma alata*. (f) Cell of an unidentified *Amphidoma***
**sp. 1 in ventral view. Note the long apical plates, the ventral depression (white arrow in f), and the row of pore on the**
**posterior suture of apical plates. (g, h). Two cells of *Amphidoma* sp. resembling *Amphidoma trioculata*. (i) Hypotheca in**
**antapical view of an unidentified *Amphidoma*. Note the multiple pores on the plates and the presence of a very small**
**antapical pores on the second antapical plate (white arrow in i). Scale bars = 2 µm.**

Due to the very similar size of *Az. dexteroporum*, *Am. parvula*, and *Am. languida*, all three species as identified with SEM are
present in the one category of small amphidomatcean cells used for light microscopic analysis and quantification.
**A 1.5. Other Amphidomataceae**
In the SEM, a few other cells were observed, which can also be assigned to the genus *Amphidoma*. The epitheca found in
dorsal view in Fig. A09e with its characteristic pore ridges on the large apical plates corresponds to *Am. alata*, a species which
was described from the Argentina shelf (Tillmann, 2018). The cell depicted in Fig. A09f in ventral view likely represents a
new species of *Amphidoma*. The cells in Figs. A09g-i likely correspond to *Am. trioculata*, another species described from
Argentina (Tillmann, 2018), though assigning the isolated hypotheca (Fig. A09i) is difficult.



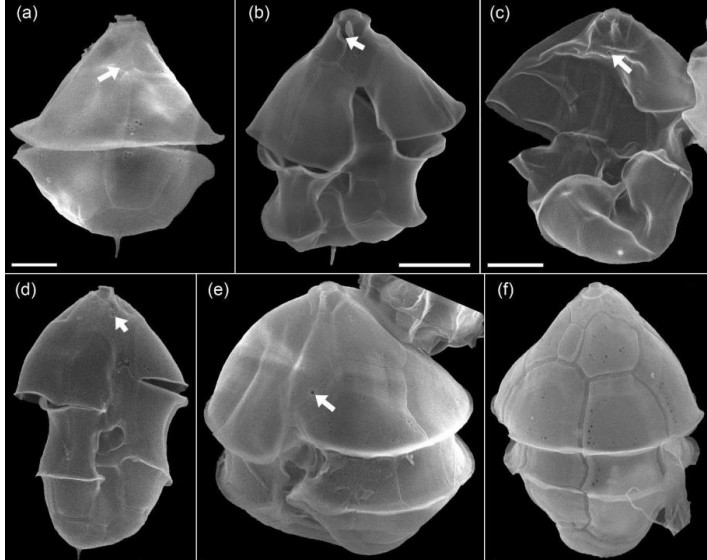


**Figure A10: SEM of unidentified cells of *Azadinium* spp. of the Amphidomatacean bloom stations. (a) Cell in dorsal**
**view. Note the very small and 5-sided intercalary plate (white arrow in a). (b) Cell of *Azadinium* sp. 1 in ventral view.**
**Note the very short 1' Plate and the position of the ventral pore inside the pore plate (white arrow in b). It may be**
**assumed but is not clear if (a) is the dorsal view of such an *Azadinium* sp. 1. (c. d), Two cells in ventral view of *Azadinium***
**sp. 2. Note the position of the ventral pore in apical position inside of Plate 1' (white arrows in c, d). (e) Cell of *Azadinium***
**sp. 3 in lateral ventral view. Note the position of the ventral pore (white arrow in e) and the rather long apical plates.**
**(f). Dorsal view of an *Azadinium* sp. 4 resembling in size and shape *Azadinium asperum*. Scale bars = 2 µm.**

Moreover, for a more complete description of the diversity of Amphidomataceae in the bloom sample, the following individual
findings (Fig. A10) should also be mentioned: A cell in dorsal view (Fig. A10a) had a very distinct antapical spine, relatively
large apical plates, and a small six-sided central intercalary plate. This combination of features has not been described in any
known *Azadinium* species, suggesting that this may represent a new species. The cell depicted in Fig. A10b had a distinct
antapical spine, a ventral pore on the right side of the pore plate, and a very short first apical plate, with the anterior sulcal
plate extending far into the epitheca. Both cells in Figs. A10b and c resemble *Az. spinosum* but differ in that the ventral pore
is centrally located within Plate 1' in the apical area. They likely correspond to the cells designated as *Azadinium* sp. 3 in Fig.
14 b, c, in Tillmann (2018). The cell in Fig. A10e had a ventral pore on the right side of Plate 1' (like *Az. spinosum* and *Az.*
*obesum*), but here the lateral apical plates were significantly larger than in these species. The cell in Fig. A10f in dorsal view
in terms of size and shape might correspond to *Az. asperum* described from the Argentine shelf (Tillmann, 2018), undoubtedly
an *Azadinium* species due to the apical pore and intercalary plates, does not match any previously described species based on
size (ca 20 µm cell length) and shape. This, along with the other cells in Fig. A09 which likely represents previously
undescribed species due to the unique combination of features, highlights the great diversity of Amphidomataceae in this 2021
bloom sample.

**Appendix B**

**Species diversity based on ITS1-based metabarcoding**




**Table B01: Species detected by Amplicon Sequence Variant (ASV) reads at stations GA01 and AA09.**

| Nr | species | Genbank | Identity % | GA01 | AA09 |
|---|---|---|---|---|---|
| 1 | *Amphidoma parvula* | KY996792 | **>98** | 19 | 0 |
| 2 | *Amphidoma* sp. 1 | OQ360107 | 90-95 | 158 | 200 |
| 3 | *Amphidoma* sp. 2 | LC788745 | 90-95 | 16 | 0 |
| 5 | *Azadinium dalianese* ribotype E | LS974150 | 90-95 | 41 | 63 |
| | | | 95-98 | 33 | 58 |
| | | | **>98** | 93,775 | 98,503 |
| 6 | *Azadinium dalianense* ribotype B | MF033117 | 95-98 | 9 | 0 |
| | | | **>98** | 0 | 19 |
| 7 | *Azadinium* sp. 1 | OQ360091 | 90-95 | 36 | 41 |
| 8 | *Azadinium* sp. 2 | OQ360094 | **>98** | 6 | 5 |
| 9 | *Azadinium spinosum* ribotype C | MK405512 | **>98** | 17,006 | 7,589 |
| 10 | *Azadinium spinosum* ribotype B | LS974169 | **>98** | 0 | 37 |
| 11 | *Ansanella* sp. | MN604385 | 90-95 | 4 | 13 |
| 12 | *Bicheleria* sp. | KC895487 | 90-95 | 12 | 52 |
| 13 | *Bicheleria cincta* | KC895487 | **>98** | 6 | 17 |
| 14 | *Blastodinium oviforme* | JX473680 | 95-98 | 2 | 0 |
| 15 | *Karlodinium decipiens* | LC521288 | 95-98 | 0 | 7 |
| | | | **>98** | 885 | 106 |
| 16 | *Karlodinium digitatum* | MN133932 | **>98** | 3 | 0 |
| 17 | *Kirithra asteri* | MW267275 | **>98** | 41 | 186 |
| 18 | *Pelagodinium beii* | KP843723 | **>98** | 48 | 77 |

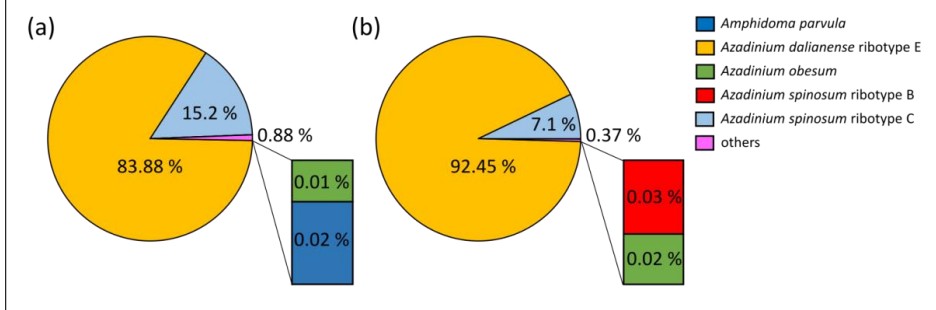

**Figure B01: Relative abundance (in %) of the Amphidomataceae species detected with metabarcoding targeting ITS1 regions at stations GA01 (a) and AA09 (b).**

**Appendix C**

**Overall appearance of the pure Amphidomataceae bloom in fixed field samples**



**Fig. C01: Micrographs of the Amphidomataceae bloom taken under light microscopy at low magnification: 200x. Only (e) was taken under 400x and (p) was taken using fluorescence in 100x. The toxic species *Azadinium spinosum* ribotype B is indicated in a red circle. The arrows indicate other protists: *Og*: *Oxytoxum gracile*, K-type: Kareniacea-type cell, *K*: *Katodinium* sp., *T*: *Tripos* sp., C: ciliate with a cell of Amphidomataceae inside, *Mr*: *Mesodinium rubrum*.**




**Appendix D**

**Retention of particles in the blooming area**

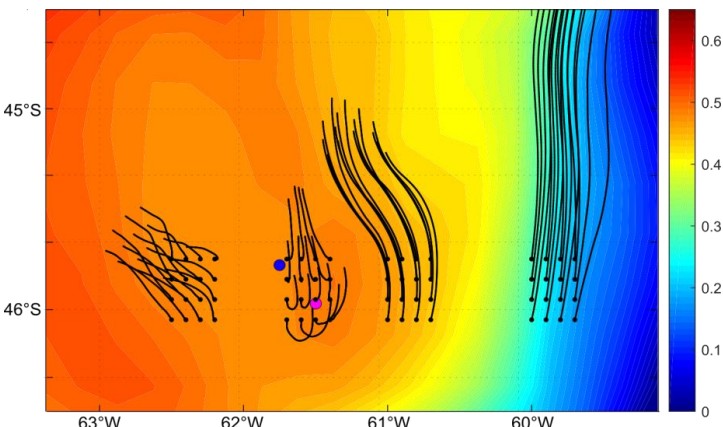


**Fig. D01: Background colours: Absolute Dynamic Topography (m) averaged from 16 to 25 November. Black lines**
**correspond to the advection of particles for the same period after release at the points indicated with a black dot on 16**
**of November.**

**Supplementary material:** Video showing the daily evolution from 10 to 25 November 2021 of the Finite-size Lyapunov
Exponent (FSLE) ridges in the area of the two locations with the Amphidomataceae bloom: GA01, pink dot, sampled on
November 16, and AA09, blue dot, sampled on November 25. The two stations remained within the same water mass separated
by two maxima FSLE.

**Data availability:** The CTD data and the abundance of protistan species counted under light microscopy at the sampling
station GA01 and AA09, will be publicly available in the repository Pangaea Data Publisher (www.pangaea.de)**.** Data from
the Gayoso cruise is available at: https://doi.pangaea.de/10.1594/PANGAEA.971564. Sequences obtained in this study are
available in the National Center for Biotechnology Information, Sequence Read Archive (http://www.ebi.ac.uk/ena).

**Author Contribution:** VAG conceptualised and designed the study, coordinated the planning and execution of field and
laboratory work, and secured funding. VAG and UT analysed the plankton samples using LM and SEM. MR processed the
satellite-derived chlorophyll data and the Lyapunov coefficients. CF and FR conducted field research and processed the CTD
data. VAG and BK processed the toxin samples. HG processed the DNA samples. MS analysed the geostrophic currents and
performed the particle tracking modelling. All co-authors contributed to the interpretation of the results. VAG prepared the
manuscript with contributions from all co-authors.

**Competing interests:** The authors declare that they have no conflict of interest.

**Acknowledgments and Funding:** VAG and MS acknowledge the BioMMAr consortium and the common grant received
from the Argentinean Initiative Pampa Azul (PIDT A6) to carry out the Gayoso cruise. VAG acknowledges the academic
mobility projects: FitoxNorPat, CONICET-DAAD call-2020, and Coastcarb, H2020-Marie Skłodowska-Curie Actions,
MSCA-RISE-2019, N° 872690. VAG especially thanks Marcelo Acha from INIDEP, Argentina, for his invitation to
participate in the WG of the Marine Priority Area of Agujero Azul, from the Pampa Azul Initiative. VAG also thanks Annegret



Mueller and Thomas Max for their help and guidance in the laboratory work on processing toxin samples, UT and HG thanks
Annegret Müller for DNA extraction. The authors thank the Argentinean authorities, the captain and crew of the RV Bernardo
Houssay (Prefectura Naval Argentina) and the RV Victor Angelescu (INIDEP).

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
