# Peer review of "Extraordinary bloom of toxin-producing phytoplankton enhanced by"

_EGUsphere, 2024_

## Referee Comment (RC1)

**General Comments**

Your study explores the biophysical drivers of harmful algal blooms (HABs) using datasets collected during two research cruises on the Patagonian shelf and satellite-based analysis. I find this topic highly interesting and am pleased to assist you in showcasing its significance.

I find the paper difficult to read due to redundant paragraphs, a disorganized structure, and many long sentences. I recommend a major revision of the article's structure to improve readability and better emphasize the key questions and conclusions, which are currently unclear. I suggest providing a clearer description of the current limitations regarding the biophysical coupling responsible for HABs, the scientific question being addressed, and how your work advances your understanding beyond the limitations. Remember that every sentence must have a clear purpose, so words must be chosen with meaning and precision.

Note that I will not provide scientific feedback on the genetic and microscopic methods and analyses, as these are outside my areas of expertise.

Although English is not my first language, I have noticed several errors. I suggest a careful revision of the spelling and syntax.

**1) Introduction**

The structure should be reworked to improve its flow by organizing the paragraphs in a logical progression from general to specific. I recommend the following steps:

- **General context:** Begin by discussing HABs in general, their ecological and socio-economic impacts, and their relationship with the physical environment, without introducing the Patagonian shelf, as that will be covered in a subsequent paragraph. Provide a thorough overview of the current state of knowledge on HABs, supported by more citations. For example, you could move the paragraph starting at line 73, "The most conspicuous HABs are those formed..." and lines 91-94, "In oceanic waters…" as it provides a general perspective that is not specific to the Patagonian shelf. This will help establish a broader context before narrowing the focus to your specific study area.

Potential references:
Pitcher et al., 2010 https://doi.org/10.1016/j.pocean.2010.02.002
Ralston & Moore, 2020 https://doi.org/10.1016/j.hal.2019.101729
Wells et al., 2020 https://doi.org/10.1016/j.hal.2019.101632
Smayda, 1997 https://doi.org/10.4319/lo.1997.42.5_part_2.1137
Iriarte et al., 2023 https://doi.org/10.1016/j.pocean.2023.103087

- **Limitations and frontiers to address**: Clearly outline the current limitations in the study of HABs, emphasizing the challenges in understanding the biophysical processes driving them. Highlight the existing knowledge gaps that hinder a comprehensive understanding of these phenomena. Explain what remains unresolved and why these questions are significant. Finally, articulate the key objectives of your study, specifying what aspects of HABs you aim to uncover and how your work seeks to advance the field. For the moment, the limitations are diluted throughout the introduction when they should be

in the same place. For example, you mention at the end, lines 93-96, "...with few studies considering in situ sampling".

- **Specific question of your paper:** Clearly state the specific research question your paper addresses. This is also the appropriate place to introduce your study region—the Patagonian shelf—and explain why understanding the processes controlling HABs in this area is important. For instance, you could emphasize the significance of the Patagonian shelf by highlighting the statement from lines 77–78: "The maximum bloom abundances reported in the literature are from the Argentine Sea." This key point should appear earlier in the paragraph to provide strong justification for your focus on this region. Currently, you present the specific question at the end of your introduction, in the lines "Furthermore, we aim …," whereas it is expected to appear earlier, before explaining your strategy.

- **Hypothesis:** Clearly state your hypothesis related to the specific research question. For example, you could propose, "Blooms within the Patagonian shelf are driven by a strong synergy between mesoscale processes and dinoflagellate communities." Avoid repeating details about the physical characteristics of the Patagonian shelf or the importance of dinoflagellates, as these should already be covered in the preceding paragraph. Focus instead on presenting a concise and well-defined hypothesis to guide readers into the objectives and analysis of your study.

- **Your strategy:** Concisely describe the approach you implemented to address your specific research question. Focus on outlining the key methods and steps taken to achieve your objectives, providing a clear and logical connection to the hypothesis and study goals.

**2) Materials and Methods**

In general, I recommend a thorough revision of this section to better describe the methods, particularly by simplifying the sentences and reorganizing the sections. Additionally, some descriptions are missing (see Specific Comments below).

I suggest removing Section 2.1, as it is somewhat redundant with the introduction. Instead, you could synthesize additional informations and incorporate it into the section of the introduction that discusses the study region.

Next, I recommend presenting the cruise details with more emphasis on the strategy, such as how the station locations were selected, the duration of each station deployment, and the number of CTD casts performed... Following that, the in situ measurements (Section 2.4) should be presented immediately after the cruise description. I also wonder if it is necessary to have two separate sections for this.

Subsequently, the satellite products (Section 2.3) should follow, with the Lagrangian trajectories and FSLEs analysis after that. Sections 2.6 and 2.7 could be consolidated into the same section as the satellite-based observations, possibly as subsections. For example, Section 2.3 could be titled "Satellite-based observations and analysis," with subsections such as 2.3.1 "Remote Sensing of Surface Chla, SST, and ADT" and 2.3.2 "Lagrangian trajectories and FSLEs analysis".

**3) Results**

Similar to the introduction and M&M sections, I suggest improving the structure of the Results section. Start by describing the study region physically using satellite-based observations and analyses. Section 3.4 should be placed earlier in the Results section, as it would allow for a more direct comparison of chlorophyll-a surface distribution with surface physical dynamics. Following this, present the in situ dataset to integrate biological and physical data, which is the central focus of your article. I also have some suggestions regarding the figures (see Specific Comments below).

Moreover, I find that the description of the results is sometimes redundant with the M&M section or the Discussion. You should aim to present the results more concisely, avoiding repetition or interpretation (see Specific Comments below).

*4) Discussion*

I suggest adding a brief summary before Section 4.1 to recap the key points of your work.

The Discussion should clearly emphasize how you address with your results the specific question outlined in the introduction. Currently, I find the Discussion somewhat confusing, and it is difficult to identify the key conclusions of your study, primarily due to long sentences, lack of organization, and redundant descriptions with the Introduction. Specifically, I believe there is an excessive focus on describing other studies, with limited explanations of your own results. I recommend synthesizing the findings from other studies to better highlight your own contributions (particularly for sections 4.1 and 4.2).

I find the proposal of different scenarios (Fig. 12) interesting, but I was expecting more discussion of them in relation to your results. Specifically, the Lagrangian simulations are not discussed at all.

I suggest adding a short paragraph at the end of the Discussion to address the limitations of your study. For example, is two stations sufficient to answer the research question? What additional analyses or strategies could be employed in the future to study the biophysical causes of HABs (e.g., a Lagrangian in situ strategy to track these blooms in space and time)?

Finally, I recommend adding a Conclusion section from line 544 to the end. However, you should rework the text to avoid simply listing the "factors".

**Specific Comments**

1)

- In the Methods section, you should provide sufficient information to allow readers to reproduce your work. For instance, if you used equations (e.g., for estimating chlorophyll-a concentrations), you need to specify the parameters used to parameterize these equations.

- Additionally, include key details for each method. For example, the description of nutrient measurements (lines 187–189) is unclear. Does the method have a specific name? What is the underlying principle? What parameters were used? Similarly, for the "screened mass transitions and instrument parameters" (lines 231–232), the explanation is vague and lacks clarity—what does this refer to, and how was it performed?

- For the Lagrangian simulations, you also need to provide more details, such as the timestep, initial conditions, and a small description of the algorithm's principle. The current sentence, "The algorithm computes the particle positions based on initial location and knowledge of the velocity field" (lines 239–240), is too brief. Similarly, in Section 2.7 (which could be combined with the preceding section since FSLEs are also part of Lagrangian analysis), you should begin with a clear definition of FSLEs and provide detailed information on how they were calculated. I strongly recommend combining the Lagrangian analysis with the satellite data section, as they are closely connected. This would prevent readers from having to flip back to earlier sections to recall where the data originated.

- In Section 2.6, the concept of ribotypes is not clear. I suggest adding a brief definition to clarify their purpose and how identifying them contributes to the study. Furthermore, the entire paragraph in this section requires a more detailed explanation, as it is currently unclear what the goal of this analysis is. It might also be worth considering combining this section with Section 2.5, as they appear to be complementary.

2) Regarding the figures, I suggest some modifications to improve their clarity.

- I suggest adding the vertical profiles of fluorescence to Figure 1. In the text, you state, "Surface water temperature and salinity remained similar at both stations GA01 and AA09," but this does not appear to be the case for temperature based on Figure 1 (~9°C for GA01 and ~11°C for AA09). It is also inconsistent with the values reported in Table 1.

- I recommend using a different color palette for Figures 2, 3, 8, 9, and D01, as the rainbow colormap is no longer widely used for mapping (see https://doi.org/10.5194/hess-25-4549-2021 and https://doi.org/10.5670/oceanog.2016.66). Additionally, the current colormap makes it difficult to distinguish the station dots, especially in Figure 9.

- For Figures 8 and 9, you should include a colorbar label with units. In the legend of Figure 9, you should specify that the pink color represents the particle trajectories and indicated the initial positions of each particle with dots or crosses.

- For figures 2, 3, 8 and 9 it is confusing to use the same colormap to represent different types of data. I recommend using distinct colormaps for different variables (e.g., http://dx.doi.org/10.1029/2018JC014392). This would help differentiate between datasets and make the figures more intuitive for readers. Furthermore, why are the longitude and latitude limits not consistent across these maps (also with Figures 10 and 11)?

- I find the purpose of Figure 11 unclear. In the text, it is mentioned in just one sentence (lines 411–412), yet the figure is complex and takes up a significant amount of space. I suggest removing Figure 11, as the same information is already presented in Figure 10. However, you could modify Figure 10 to include, in addition to the large map, a zoomed-in view of the sampling stations (using only the inset subplot in pink from Figure 11.

3) My main concern about your work is the robustness and originality of the in situ observations.

- Having only two stations is not an issue for me; I understand the challenges of collecting in situ data, and I believe your discussion appropriately reflects the scope of your dataset without overextending the

conclusions. However, I could not find informations about the number of CTD casts, the number of replicates, or the timing of observations at stations GA01 and AA09 (e.g., were they conducted during the day or at night?). These details are essential for assessing the reliability of your results.

- Additionally, why were samples collected only at a depth of 5m? Was the bloom detected during the cruise or afterward? Were the locations of GA01 and AA09 specifically chosen for this reason? You should improve the section describing the cruise strategy (refer to my comments in point 2 of the General Comments) to provide more comprehensive detail and highlight why your data are both robust and original.

4) I wonder if the taxonomic composition of the bloom at stations GA01 and AA09 is the same as at the other stations. Did you compare the composition of these two stations with that of the others? Highlighting such a comparison could emphasize why this bloom is extraordinary. You briefly mentioned this in the Discussion (lines 537 to 540), but I believe it should be given more prominence. Instead of including it in this section of the Discussion, it would be better placed in the section where you describe the biological characteristics of the bloom.

5) I expected more discussion about the role of the frontal system, particularly in the final section of the Discussion. How does the presence of the front influence this bloom? In my view, the front acts as a hydrodynamic barrier, preventing the dispersion of the bloom. Additionally, smaller-scale physical phenomena, such as mesoscale and submesoscale eddies, also play a significant role in the bloom's behavior through horizontal stirring and retention processes. Your sentences on lines 509–511 and lines 515–518 are not very clear and could benefit from further clarification.

6) I like your final figure; however, why are you including the last scenario, "independent bloom patch," if it is not likely? This figure should focus solely on the plausible explanations for your results.

**Technical corrections**

In general, be careful to keep your sentences shorter and to streamline your text by avoiding excessive use of extra words like "indeed," "for instance," etc… Moreover, ensure consistency in your notation style throughout the manuscript.

1) Lines 30 to 32: "The magnitude of this bloom is a global record for this group so far reported in the literature. The toxin azaspiracid-2 **[add "(AZA-2)"]** was detected in both stages of the bloom, with values up to 2122 pg L-1. The most likely source of AZA-2 was Azadinium spinosum ribotype B."

2) Sentence lines 33 to 35: "Water retention..." is too long.

3) lines 46: "Dinoflagellates produce a wide range of toxins" is redundant with the previous sentence.

4) Sentence lines 48 to 50: "In the Argentina..." is too long.

5) Line 52: "as documented for instance" to remove.

6) Line 56: "indeed" to remove.

7) Line 58 :"Furthemore" to remove.

8) Line 61: replace "important hazards" by "unexpectedly" or something like that

9) Line 68: "T**h**ermohaline"

10) Line 70: "Additionally" to remove.

11) Line 79: "Liter" and not "Litre". I noticed this mistake several times in the manuscript.

12) Line 81: "in the area" to remove.

13) Line 83: "Meter" and not "Metre". I aslo noticed this mistake several times in the manuscript.

14) Sentence line 93 to 97: "Typically…" is too long.

15) Line 111: "The steep slope", what are you talking about?

16) Sentence line 123 to 126: "Hence,…" is too long.

17) Sentence lines 137 to 138: "We first..." can be removed.

18) Sentence lines 151 to 152: "In order to" can be removed. Therefore, you will need to reformulate the next sentence.

19) Be careful to maintain consistency in your notations. For example, on line 157, you used a "-" between dates, whereas on line 158, you used "to"

20) Similarly, units should be presented in a consistent short format throughout. For example, on line 177, change "in meter" to "m" to align with line 179, where "5 m depth" is used.

21) Line 182: change "for 4h" by "during 4h".

22) Line 204: a space is missing before "A1".

23) Line 205: "Thereafter, they were…", who are "they"?

24) Line 206: "Further" to remove.

25) Sentence lines 236 to 237 "We used…" can be removed.

26) Line 261: "a**n** uniform…"

27) Figure 2: figure letters a), b), c), d) are missing.

28) Lines 271 to 272: "On November.." where do you see that?

29) Line 309: "A reddish water discoloration…" where do you see that?

30) Figure 4: "around 60m" it is more like 50m.

31) Line 334: "ITS", what is it?

32) Sentence lines 339 to 340: "This distinction..." not useful as it should be in the method section.

33) Figures 6 and 7: either "scale bar = …" or "scale bar is …".

34) Figures 2, 6 and 7: either a) or (a). Be consistent.

35) Line 373: "Lyapunov frontal systems" means nothing. I suggest to reformulate the title of this section as "Description of the frontal systems" or something like that.

36) Line 386: "80 cm/s" while in the figure it is in m/s.

37) Line 408: "Moreover" to remove. And "exponent**s** (FSLE**s**)"

38) Figure 10: modify the caption as following or something like that "FSLEs fields computed…"

39) Lines 462-463: "And in particular" to remove.

40) Line 486: "In fact" to remove.

41) Line 533: "Additionally" to remove.

42) Sentence lines 540 to 542: "In this study…", is incorrect. Retention is an in situ process and cannot be constrained by satellite observations.

43) Lines 548 to 558: Reorganize the sentences to avoid using a listing format.

---

## Author Comment (AC1)

**RC1**: 'Comment on egusphere-2024-3157', Anonymous Referee #1, 20 Dec 2024 reply

Please refer to the PDF for more details regarding all the following comments.

1. **Does the paper present novel concepts, ideas, tools, or data?**

   Partially. While the paper presents a bloom that has not been previously reported, the novelty and value of the dataset could be better emphasized. Highlighting the unique aspects of cruises strategy would underline the distinctiveness of the data collected.

2. **Are substantial conclusions reached?**

   Yes, but the discussion section needs improvement to better articulate the connection with the physical seascape.

3. **Are the scientific methods and assumptions valid and clearly outlined?**

   No. The authors need to clearly state their hypotheses in the introduction and then outline the methods used to test them.

4. **Are the results sufficient to support the interpretations and conclusions?**

   Yes, I think they are, but reworking the manuscript will make this link clearer and more convincing. Particularly, by discussing more about the story tell by Lagrangian simulations.

5. **Is the description of experiments and calculations sufficiently complete and precise to allow their reproduction by fellow scientists (traceability of results)?**

   No. The methods section requires more detailed descriptions about the strategy of cruises and methods.

6. **Do the authors give proper credit to related work and clearly indicate their own new/original contribution?**

   No. While I believe this work is original as mentioned earlier, reworking the manuscript would help clearly highlight that.

7. **Does the title clearly reflect the contents of the paper?**

   Yes, but the retention process should be described in more detail within the discussion to align fully with the title's focus.

8. **Does the abstract provide a concise and complete summary?**

   Yes.

9. **Is the overall presentation well structured and clear?**

Not entirely. While the paper has valuable content, there are some spelling errors, inconsistencies (e.g., units, figures), and long sentences. Additionally, the structure could be improved to enhance clarity and coherence.

10. **Is the language fluent and precise?**

No. See pdf.

11. **Are mathematical formulae, symbols, abbreviations, and units correctly defined and used?**

No. See pdf.

12. **Should any parts of the paper (text, formulae, figures, tables) be clarified, reduced, combined, or eliminated?**

No. See pdf.

13. **Are the number and quality of references appropriate?**

No. See pdf.

14. **Is the amount and quality of supplementary material appropriate?**

Yes.

Author reply: Thank you very much for your comments and constructive feedback. We have followed your suggestions to improve structuring and we have made modifications to emphasise the physical seascape of our original results.

**General Comments**

Your study explores the biophysical drivers of harmful algal blooms (HABs) using datasets collected during two research cruises on the Patagonian shelf and satellite-based analysis. I find this topic highly interesting and am pleased to assist you in showcasing its significance. I find the paper difficult to read due to redundant paragraphs, a disorganized structure, and many long sentences. I recommend a major revision of the article's structure to improve readability and better emphasize the key questions and conclusions, which are currently unclear. I suggest providing a clearer description of the current limitations regarding the biophysical coupling responsible for HABs, the scientific question being addressed, and how your work advances your understanding beyond the limitations. Remember that every sentence must have a clear purpose, so words must be chosen with meaning and precision. Note that I will not provide scientific feedback on the genetic and microscopic methods and analyses, as these are outside my areas of expertise.
Although English is not my first language, I have noticed several errors. I suggest a careful revision of the spelling and syntax.

Author reply: We appreciate the suggestions to help improve the clarity of our writing and emphasize our original findings. We have refined the formulation of our questions and methods

applied during the oceanographic cruises. Additionally, we have added some lines highlighting the current limitations and advancements in our understanding of the biophysical aspects involved in the development of phytoplankton blooms in offshore waters. Spelling and syntax have been revised.

**1) Introduction**

The structure should be reworked to improve its flow by organizing the paragraphs in a logical progression from general to specific. I recommend the following steps:

Author reply: We appreciate the suggestions for restructuring the Introduction. However, we have incorporated them partially, as we believe the original structure and flow are consistent with the organization of the other sections, gradually narrowing the focus from general to specific aspects of our work. We have also taken into account the positive feedback from Reviewer #2, who praised the manuscript's well-written structure.

- **General context:** Begin by discussing HABs in general, their ecological and socio-economic impacts, and their relationship with the physical environment, without introducing the Patagonian shelf, as that will be covered in a subsequent paragraph. Provide a thorough overview of the current state of knowledge on HABs, supported by more citations. For example, you could move the paragraph starting at line 73, "The most conspicuous HABs are those formed..." and lines 91-94, "In oceanic waters…" as it provides a general perspective that is not specific to the Patagonian shelf. This will help establish a broader context before narrowing the focus to your specific study area.
Potential references:
Pitcher et al., 2010 https://doi.org/10.1016/j.pocean.2010.02.002
Ralston & Moore, 2020 https://doi.org/10.1016/j.hal.2019.101729
Wells et al., 2020 https://doi.org/10.1016/j.hal.2019.101632
Smayda, 1997 https://doi.org/10.4319/lo.1997.42.5_part_2.1137
Iriarte et al., 2023 https://doi.org/10.1016/j.pocean.2023.103087

Author reply: We have moved lines 91–94 to the first paragraph, as suggested. Line 73 remains in its original position because it provides specific examples of Amphidomataceae species in the Argentine Sea. We have kept the brief mention of HABs in the Patagonian Shelf in this first paragraph because it is relevant to provide general examples of toxic events and their producers, highlighting that most HABs studies have been conducted in the coastal zones of the Argentine Sea.

We have incorporated the following three references in the revised version of the manuscript: Iriarte et al. (2023), which is a short review of HABs along the Chilean coast, now also cited in the Discussion when comparing our results with similar shelves on the Pacific coast of South America; Pitcher et al. (2010), which discusses physical drivers of HABs in upwelling systems; and Wells et al. (2020), which comments on the physical, biological, and chemical changes in the ocean driven by climate change and their potential link to HABs, as well as the challenges of studying HABs through in situ monitoring and modelling, and the need to apply multiple approaches to assess HABs diversity. Although Smayda (1997) is a comprehensive, classic review of HABs in marine waters, we have not added it because we cited other related references from the same and different authors. The work of Ralston and Moore (2020) addresses modelling of HABs under future climate change scenarios; hence, we have not included this reference because our focus is on synoptic sampling.

- **Limitations and frontiers to address**: Clearly outline the current limitations in the study of HABs, emphasizing the challenges in understanding the biophysical processes

driving them. Highlight the existing knowledge gaps that hinder a comprehensive understanding of these phenomena. Explain what remains unresolved and why these questions are significant. Finally, articulate the key objectives of your study, specifying what aspects of HABs you aim to uncover and how your work seeks to advance the field. For the moment, the limitations are diluted throughout the introduction when they should be in the same place. For example, you mention at the end, lines 93-96, "...with few studies considering in situ sampling".

Author reply: We have made some modifications in the text to better articulate the knowledge gaps that hinder the understanding of HABs and the objectives of our work.

- **Specific question of your paper:** Clearly state the specific research question your paper addresses. This is also the appropriate place to introduce your study region—the Patagonian shelf—and explain why understanding the processes controlling HABs in this area is important. For instance, you could emphasize the significance of the Patagonian shelf by highlighting the statement from lines 77–78: "The maximum bloom abundances reported in the literature are from the Argentine Sea." This key point should appear earlier in the paragraph to provide strong justification for your focus on this region.
Currently, you present the specific question at the end of your introduction, in the lines "Furthermore, we aim …," whereas it is expected to appear earlier, before explaining your strategy.

- **Hypothesis:** Clearly state your hypothesis related to the specific research question. For example, you could propose, "Blooms within the Patagonian shelf are driven by a strong synergy between mesoscale processes and dinoflagellate communities." Avoid repeating details about the physical characteristics of the Patagonian shelf or the importance of dinoflagellates, as these should already be covered in the preceding paragraph. Focus instead on presenting a concise and well-defined hypothesis to guide readers into the objectives and analysis of your study.

- **Your strategy:** Concisely describe the approach you implemented to address your specific research question. Focus on outlining the key methods and steps taken to achieve your objectives, providing a clear and logical connection to the hypothesis and study goals.

Author reply: We have taken under consideration some of the suggested changes to restructure the introduction section, but we still keep its original order for coherence with all the other sections of the manuscript. In the last paragraph, we have revised and made modifications to the formulation of our specific questions, hypothesis and our research strategy.

*2) Materials and Methods*
In general, I recommend a thorough revision of this section to better describe the methods, particularly by simplifying the sentences and reorganizing the sections. Additionally, some descriptions are missing (see Specific Comments below).
I suggest removing Section 2.1, as it is somewhat redundant with the introduction. Instead, you could synthesize additional informations and incorporate it into the section of the introduction that discusses the study region.
Next, I recommend presenting the cruise details with more emphasis on the strategy, such as how the station locations were selected, the duration of each station deployment, and the number of CTD casts performed... Following that, the in situ measurements (Section 2.4) should be presented immediately after the cruise description. I also wonder if it is necessary to have two separate sections for this.

Subsequently, the satellite products (Section 2.3) should follow, with the Lagrangian trajectories and FSLEs analysis after that. Sections 2.6 and 2.7 could be consolidated into the same section as the satellite-based observations, possibly as subsections. For example, Section 2.3 could be titled "Satellite-based observations and analysis," with subsections such as 2.3.1 "Remote Sensing of Surface Chla, SST, and ADT" and 2.3.2 "Lagrangian trajectories and FSLEs analysis".

Author reply: The organization of all the materials and methods has been carefully discussed to find the most coherent order to introduce the multiple approaches applied in our work –e.g. cruise description, remote sensing of chlorophyll-a, biogeochemical in situ measurements and water collection, species diversity by microscopy and genetic analyses, toxin profiles, surface currents and fronts – which is consistent with the Results and the Discussion sections (see the following responses regarding order and structure).

In the M&M section of the revised version of the manuscript, we have made the following modifications according to the reviewer's suggestions:

Section 2.4: this concise section is needed to explain Figure 1, and to introduce the main circulation patterns that drive the high phytoplankton productivity in the Patagonian Shelf-break front. We have combined part of this section with the Introduction in order to avoid repetition.

We have added more information regarding the sampling strategy of the two oceanographic cruises as requested. For instance, Gayoso cruise was planned mainly to analyse the pre-bloom conditions of the coccolithophore *Emiliania huxleyi* which tends to bloom in early summer (December) in the Patagonian shelf. This cruise was planned tracking daily satellite-derived signals of particulate inorganic carbon and chlorophyll-a. We have updated the reference Ferronato et al, 2025 (accepted in J. Geophys. Res. Oceans) and Gilabert et al., 2025, https://doi.org/10.1007/s10533-024-01192-6, which provide more details about the Gayoso cruise strategy and selection of stations' location (both references have been updated in the revised manuscript). Similarly, we have added more information about the sampling strategy of the Agujero Azul research expedition, which was planned as part of an interdisciplinary project to analyse the pelagic and benthic biodiversity of this area, highly important for fisheries activities: https://www.pampazul.gob.ar/investigacion-y-desarrollo/areas-prioritarias/agujero-azulfrente-del-talud/

We have moved the last paragraph of section 2.3 (Lines 165 to 169: information of the satellite images used for Lagrangian and Lyapunov analyses) to Section 2.7, where we have merged Lagrangian simulations and Finite Size Lyapunov Exponent analysis. In the revised version of the manuscript, Section 2.3 is now only Remote sensing of Chl-a and SST.

**3) Results**

Similar to the introduction and M&M sections, I suggest improving the structure of the Results section. Start by describing the study region physically using satellite-based observations and analyses. Section 3.4 should be placed earlier in the Results section, as it would allow for a more direct comparison of chlorophyll-a surface distribution with surface physical dynamics. Following this, present the in situ dataset to integrate biological and physical data, which is the central focus of your article. I also have some suggestions regarding the figures (see Specific Comments below).

Moreover, I find that the description of the results is sometimes redundant with the M&M section or the Discussion. You should aim to present the results more concisely, avoiding repetition or interpretation (see Specific Comments below).

Author reply: We have kept the original order of the subsections and figures. In response to the Reviewer's suggestion, we have revised the text and made changes to avoid repetition or misinterpretation in the Results.

The reviewer's suggestions regarding the structure of the paper were considered in such a way that the original order was preserved, highlighting the following steps and results of the study: The discovery of the same bloom persisting for at least 10 days in hydrographically complex waters near the continental slope was unprecedented in Patagonia. As we have explained more about the sampling strategy in M&M in the revised version of the ms, the sampling of this bloom was not planned in advance; the synoptic stations were strategically located with different objectives for each of the two oceanographic cruises, where monitoring of surface chlorophyll patches was a primary goal, along with scanning their biodiversity and phycotoxin profiles. This is why we later sought to understand the underlying drivers of this extraordinary bloom of Amphidomataceae, its development and retention. To do this, we first needed to confirm that it was indeed the same bloom and the same water mass, by analysing the spatio-temporal evolution of surface chlorophyll-a and the detailed species composition and abundance of phytoplankton species and phycotoxins. After conducting the biogeochemical analyses that confirmed that we had sampled the same bloom patch in the same water mass (nutrient levels, vertical structure), we proceeded to investigate the physical factors that might have favoured the development and retention of the bloom at a mesoscale.

*4) Discussion*

I suggest adding a brief summary before Section 4.1 to recap the key points of your work. The Discussion should clearly emphasize how you address with your results the specific question outlined in the introduction. Currently, I find the Discussion somewhat confusing, and it is difficult to identify the key conclusions of your study, primarily due to long sentences, lack of organization, and redundant descriptions with the Introduction. Specifically, I believe there is an excessive focus on describing other studies, with limited explanations of your own results. I recommend synthesizing the findings from other studies to better highlight your own contributions (particularly for sections 4.1 and 4.2).

Author reply: A brief summary of the key points of our work was originally placed at the end of the Discussion (from line 548). We have moved this paragraph before Section 4.1, as suggested by the Reviewer. Additionally, we have revised the spelling and syntax to improve readability. Regarding other studies, we believe that in the Discussion it is important to provide a comprehensive comparison of our results with similar studies on Amphidomatacean blooms in the global seas, in order to emphasize the originality of our findings and highlight that this bloom in the Argentine Sea is a world record.

I find the proposal of different scenarios (Fig. 12) interesting, but I was expecting more discussion of them in relation to your results. Specifically, the Lagrangian simulations are not discussed at all.

Author reply: Thank you for this comment. We expanded the discussion about the Lagrangian simulations as requested by the reviewer in this section.
The sentence "In addition to the biological evidence confirming the presence of the same Amphidomataceae bloom at both sampling stages, analyses of circulation through altimetry, particle experiments, and FSLEs—an indicator of frontal activity and stirring intensity—support the conclusion that the same bloom patch was captured at both locations."

Now reads: "In addition to the biological evidence confirming the presence of the same Amphidomataceae bloom at both sampling stages, analyses of circulation through satellite altimetry showed that an anticyclone of about 100 km of diameter was the responsible feature to retain the bloom within the two sampling stations. The retention that the eddy caused was evidenced by doing

two Lagrangian experiments. In the first one, the trajectories of virtual particles released within the eddy show that almost none of the particles escaped from the eddy during the 10 days that separated the two sampling stations. In the second one, FSLEs maps showed that no fronts separated the two sampling stations during those days."

I suggest adding a short paragraph at the end of the Discussion to address the limitations of your study. For example, is two stations sufficient to answer the research question? What additional analyses or strategies could be employed in the future to study the biophysical causes of HABs (e.g., a Lagrangian in situ strategy to track these blooms in space and time)? Finally, I recommend adding a Conclusion section from line 544 to the end. However, you should rework the text to avoid simply listing the "factors".

Author reply: We have added a brief paragraph as final remarks at the end of the Discussion to address gaps in our understanding of HABs, the limitations of our study, and future recommendations for tracking short-lived toxic blooms in offshore waters of the Patagonian Shelf.

The Lagrangian experiments performed in this work with virtual particles were very useful to show how a bloom of HABs trapped within an eddy can be enhanced and persist longer than if the bloom occurred elsewhere. Lagrangian experiments are possible also in the real ocean, not virtually. For example, in the framework of the TARA Microbiome Mission Gayoso-Patagonia cruise on December 2021, the vessel performed several shallow CTD stations to measure physical and biogeochemical properties following a triplet of surface drifters anchored at 15 m depth during almost four days (Ibarbalz et al., under preparation). Another possibility is to release inherent tracers and follow them (Archer et al., 2002, https://doi.org/10.1016/S0967-0645(02)00067-X). This kind of experiments will allow to obtain multiple samples while following the same water mass, and thus overcome one of the main limitations of this study, ie to have only two stations with water samples.

**Specific Comments**

1)
- In the Methods section, you should provide sufficient information to allow readers to reproduce your work. For instance, if you used equations (e.g., for estimating chlorophyll-a concentrations), you need to specify the parameters used to parameterize these equations.
- Additionally, include key details for each method. For example, the description of nutrient measurements (lines 187–189) is unclear. Does the method have a specific name? What is the underlying principle? What parameters were used? Similarly, for the "screened mass transitions and instrument parameters" (lines 231–232), the explanation is vague and lacks clarity—what does this refer to, and how was it performed?
Author reply: The materials and methods for nutrients, chl-a, and AZAs were applied following standard protocols, which are described in detail in the corresponding references cited in the text, ensuring the reproducibility of the techniques.

- For the Lagrangian simulations, you also need to provide more details, such as the timestep, initial conditions, and a small description of the algorithm's principle. The current sentence, "The algorithm computes the particle positions based on initial location and knowledge of the velocity field" (lines 239–240), is too brief.
Similarly, in Section 2.7 (which could be combined with the preceding section since FSLEs are also part of Lagrangian analysis), you should begin with a clear definition of FSLEs and provide detailed information on how they were calculated.

I strongly recommend combining the Lagrangian analysis with the satellite data section, as they are closely connected. This would prevent readers from having to flip back to earlier sections to recall where the data originated.

Author reply: All suggestions have been incorporated. We have merged sections 2.6 (Lagrangian) and 2.7 (FSLEs), and we have kept all the satellite Chl-a data in Section 2.3.

The new merged Section reads as follows in the revised version of the manuscript:

**2.7 Lagrangian simulations and Finite Size Lyapunov Exponent analysis**

To explore the physical mechanisms that might explain the concentration of amphidomatacean measured in the two locations sampled we used two complementary analysis: Lagrangian advection of virtual particles and Finite Size Lyapunov Exponents (FSLEs).

The first technique consists on the analysis of trajectories of virtual neutrally buoyant particles that were obtained with an algorithm that represents the advection process caused by surface currents. The advection equation:

$$X(t + \Delta t) = X(t) + \int_{t}^{t+\Delta t} v(x, \tau)\, d\tau \qquad (1)$$

where X is the three-dimensional position of a particle, $v(x, \tau)$ is the three-dimensional velocity field, is integrated using a fourth-order Runge–Kutta scheme. Particles were released at the surface along 46°S and every 0.05 grades in the four regions indicated in the **Appendix D**. The algorithm computes the particle positions based on initial location and knowledge of the velocity field. A time step of one hour was considered. The accuracy of the trajectories obtained relies on the accuracy of the velocity field used. For this experiment we considered geostrophic velocities obtained from satellite altimetry. In the northern portion of the Argentine continental shelf, such surface velocities showed to be well correlated with in situ current measurements (Lago et al., 2021). We therefore assume that the surface dynamics can be represented by satellite altimetry derived data and use it as the input velocity field for the algorithm to advect the virtual particles. Geostrophic velocities derived from gridded Absolute Dynamic Topography (ADT) of daily temporal resolution and ¼ of degree spatial resolution maps were downloaded from CMEMS (https://marine.copernicus.eu/). FSLE images with a spatial resolution of 1/25° grid were downloaded from AVISO (https://www.aviso.altimetry.fr).

The second technique consists on the analysis of FSLEs images with a spatial resolution of 1/25° grid that were obtained from AVISO (https://www.aviso.altimetry.fr). FSLEs ridges approximate the so-called Lagrangian Coherent structures which are the generalization of stable hyperbolic trajectories of time independent flow. They are defined as the larger eigenvalues of the Cauchy-Green strain tensor of the flow map. FSLEs are strongly linked with the exponential rate λ of separation of two neighbouring particles during a time advection t:

$$\lambda = t^{-1} \log(\delta f/\delta 0) \qquad (2)$$

where δ0 and δf are the initial and final separation distance which are fixed before computation. FSLEs are commonly used as an indicator of frontal activity and stirring intensity (d'Ovidio et al., 2004). Relatively large FSLEs values are associated with formerly distant water masses, whose confluence creates a transport front (d'Ovidio et al., 2004; d'Ovidio et al., 2009). Fronts identified as maxima (ridges) of FSLEs have a convergent dynamics transverse to them, so that passive particles in their neighbourhood are attracted to the front and then advected along it (Della Penna et al., 2015). In order to examine meso- and submesoscale frontal structures during phytoplankton blooms, daily FSLEs images from November 10 to 25 were analysed. The daily images were used to create a video (**Appendix D**) to illustrate the daily evolution of the FSLEs in the area where the phytoplankton bloom developed.

- In Section 2.6, the concept of ribotypes is not clear. I suggest adding a brief definition to clarify their purpose and how identifying them contributes to the study. Furthermore, the entire paragraph in this section requires a more detailed explanation, as it is currently unclear what the goal of this analysis is.

It might also be worth considering combining this section with Section 2.5, as they appear to be complementary.

Author reply: Although they are indeed complementary analyses to assess species-specific diversity in water samples of plankton, both microscopy and genetic techniques are different laboratory approaches. The first one is mainly based on analysing morphological aspects of the cells using microscopy, the second one is based on the molecular information to identify species. We have merged both sections as suggested by the reviewer, now called Plankton diversity analyses. We believe that the purpose of applying genetic analysis is well-stated in our work ("….was used as a complement to the exhaustive morphological taxonomy performed under light microscopy and SEM"). Ribotypes (a strain-specific feature of an organism based on variation in ribosomal RNA) are important to identify for number of Amphidomataceae species (including Az. spinosum), as within this species ribotype, specific differences in toxin production and toxin profile have been described from plankton samples collected in the Argentine Patagonian Shelf (Tillmann et al., 2019; https://doi.org/10.1016/j.hal.2019.01.008). This information is now added to this paragraph as well the reference to Appendix B.

2) Regarding the figures, I suggest some modifications to improve their clarity.
- I suggest adding the vertical profiles of fluorescence to Figure 1. In the text, you state, "Surface water temperature and salinity remained similar at both stations GA01 and AA09," but this does not appear to be the case for temperature based on Figure 1 (~9°C for GA01 and ~11°C for AA09). It is also inconsistent with the values reported in Table 1.

Author reply: Figure 1 is already a complex figure with various panels and information, so we have kept it in its original form. The vertical fluorescence profiles, along with all the CTD and biological data from our study, will be made available in a public repository once our work is published, as requested by the Journal. Data from the Gayoso cruise can be found at the following link https://doi.pangaea.de/10.1594/PANGAEA.971564 (Line 788). Thank you for pointing out the error in the SST at station GA01 in Figure 4. The 10.5°C value is correct in the table, and in the revised version of the manuscript, we have corrected the figure accordingly.

- I recommend using a different color palette for Figures 2, 3, 8, 9, and D01, as the rainbow colormap is no longer widely used for mapping (see https://doi.org/10.5194/hess-25-4549-2021 and https://doi.org/10.5670/oceanog.2016.66). Additionally, the current colormap makes it difficult to distinguish the station dots, especially in Figure 9.

Author reply: We appreciate your suggestion and the references provided. We acknowledge that color-blind friendly palettes, such as Viridis or Plasma, are becoming increasingly common in marine science following these publications. However, we have chosen to retain the current color palette for Figures 2, 3, 8, 9, and D01 because we have carefully considered its use in relation to similar studies and the visualization of key features. For the ADT and SST figures, we still believe that the current colormap effectively highlights the features we wish to emphasize. Additionally, for chl-a, we have kept the color palette used by NASA, which is widely accepted for interpreting this phytoplankton biomass proxy.

Regarding the station dots, we tested several color options and selected the most contrasting ones for clarity across all figures, ensuring consistency for stations from both cruises. In the revised version of the manuscript, we have also thickened the black outline of the station dots to enhance their contrast with the background.

- For Figures 8 and 9, you should include a colorbar label with units. In the legend of Figure 9, you should specify that the pink color represents the particle trajectories and indicated the initial positions of each particle with dots or crosses.

Author reply: We have included a colorbar label with units in Figures 8 and 9. We have also specified that the pink color represents the particle trajectories. The initial positions are indicated In Materials and Methods, the particles were released along the 46°S.

- For figures 2, 3, 8 and 9 it is confusing to use the same colormap to represent different types of data. I recommend using distinct colormaps for different variables (e.g., http://dx.doi.org/10.1029/2018JC014392). This would help differentiate between datasets and make the figures more intuitive for readers. Furthermore, why are the longitude and latitude limits not consistent across these maps (also with Figures 10 and 11)?

Author reply: We selected the current colors after testing different styles for better visualization. The use of similar color palettes for different variables should not cause confusion, as the figures are presented in a coherent order with the text, and each figure includes a detailed legend and a colorbar label with units in the revised version of the manuscript.

Regarding the variance in the longitude and latitude limits of some figures, we selected the optimal visualization for each specific result and its discussion. Figures 3 and 11 cover the same domain, they offer a zoomed-in view of the sampling stations to illustrate the daily evolution of surface chlorophyll and surface FSLEs, respectively. Figures 8 and 9 display the Lagrangian simulations in the study area. Figure 8 shows contrasting results over the shelf and the continental slope, while in Figure 9 it is important to show the velocities of the Brazil and Malvinas Currents and their confluence in the adjacent ocean basin. Figure 10 highlights the contrasting behavior of FSLEs over the shelf and in open ocean waters, including all the sampling stations from the Gayoso cruise, which are spread across the continental shelf margin, the Malvinas Current, and two stations in the highly stirred open ocean waters. All figures indicate the corresponding latitude and longitude, a scale bar showing the distance in km, and the locations of the blooming stations.

- I find the purpose of Figure 11 unclear. In the text, it is mentioned in just one sentence (lines 411–412), yet the figure is complex and takes up a significant amount of space. I suggest removing Figure 11, as the same information is already presented in Figure 10. However, you could modify Figure 10 to include, in addition to the large map, a zoomed-in view of the sampling stations (using only the inset subplot in pink from Figure 11.

Author reply: Figures 10 and 11 are substantially different. Figure 10 provides a general view of the FSLEs, showing the contrasting values in the different regions of the Southwestern Atlantic. And it does it for the 16th of November only. Figure 11 shows the evolution in time of FSLEs maps between the first and second sampling (16 and 25 of November). Figure 11 is remarkably consistent with our hypothesis: it suggests how the sampling occurred within the same water mass, as no fronts were detected in-between from the 16 to the 25 of November. It is therefore an important piece of evidence to sustain our findings.

3) My main concern about your work is the robustness and originality of the in situ observations.
- Having only two stations is not an issue for me; I understand the challenges of collecting in situ data, and I believe your discussion appropriately reflects the scope of your dataset without overextending the conclusions. However, I could not find informations about the number of CTD casts, the number of replicates, or the timing of observations at stations GA01 and AA09 (e.g., were they conducted during the day or at night?). These details are essential for assessing the reliability of your results.

- Additionally, why were samples collected only at a depth of 5m? Was the bloom detected during the cruise or afterward? Were the locations of GA01 and AA09 specifically chosen for this reason? You should improve the section describing the cruise strategy (refer to my comments in point 2 of the General Comments) to provide more comprehensive detail and highlight why your data are both robust and original.

Author reply: Please see our response to similar comments above. We have explained more the sampling strategy and added a brief Conclusion section addressing gaps and limitations of our study.

4) I wonder if the taxonomic composition of the bloom at stations GA01 and AA09 is the same as at the other stations. Did you compare the composition of these two stations with that of the others? Highlighting such a comparison could emphasize why this bloom is extraordinary. You briefly mentioned this in the Discussion (lines 537 to 540), but I believe it should be given more prominence. Instead of including it in this section of the Discussion, it would be better placed in the section where you describe the biological characteristics of the bloom.

Author reply: We have mentioned in the Discussion that this same bloom was not detected at any other sampling station of either Gayoso or Agujero Azul cruise. Some Amphidomataceae were present at other stations such as GA10 and GA09, but in very low abundance and with different species composition (see Ferronato et al. 2025). We have added this information in the revised version of the manuscript, including that no Amphidomataceae or Azaspiracids were found near station AA09 in the Agujero Azul cruise.

5) I expected more discussion about the role of the frontal system, particularly in the final section of the Discussion. How does the presence of the front influence this bloom? In my view, the front acts as a hydrodynamic barrier, preventing the dispersion of the bloom. Additionally, smaller-scale physical phenomena, such as mesoscale and submesoscale eddies, also play a significant role in the bloom's behavior through horizontal stirring and retention processes. Your sentences on lines 509–511 and lines 515–518 are not very clear and could benefit from further clarification.

Author reply: We agree with the reviewer that the front acts as a hydrodynamic barrier preventing the dispersion of the bloom. We believe that the role of frontal systems in modulating water mass movement including divergence, convergence and mixing, along with their influence on the accumulation/dispersion of plankton, is addressed in our Discussion. As previously explained, our study has limitations, and the exploration of the physical processes is at a meso-scale resolution. Therefore, we will not elaborate further on the role of submesoscale processes in modulating phytoplankton blooms, as we fear this could lead to speculation due to the lack of measurements at that scale.

6) I like your final figure; however, why are you including the last scenario, "independent bloom patch," if it is not likely? This figure should focus solely on the plausible explanations for your results.

Author reply: We are glad to hear you liked the schematic figure. We are including the last scenario because it could be a potential explanation for the bloom, although it is less likely compared to the other scenarios. We believe that presenting all the potential scenarios that might explain the same bloom observed at the two stations also enriches the discussion of the complex interplay of bio-physical drivers of phytoplankton blooms.

**Technical corrections**

In general, be careful to keep your sentences shorter and to streamline your text by avoiding excessive use of extra words like "indeed," "for instance," etc… Moreover, ensure consistency in your notation style throughout the manuscript.

Author reply: We have taken under consideration the Reviewer's suggestion in the revised version of the manuscript.

1) Lines 30 to 32: "The magnitude of this bloom is a global record for this group so far reported in the literature. The toxin azaspiracid-2 **[add** "**(AZA-2)"]** was detected in both stages of the bloom, with values up to 2122 pg L-1. The most likely source of AZA-2 was Azadinium spinosum ribotype B."
Author reply: (AZA-2) has been added as suggested.

2) Sentence lines 33 to 35: "Water retention..." is too long.
Author reply: The sentence has been shorten by splitting it into two sentences. "…shelf. This was evidenced by…"

3) lines 46: "Dinoflagellates produce a wide range of toxins" is redundant with the previous sentence.
Author reply: Checked and modified accordingly.

4) Sentence lines 48 to 50: "In the Argentina..." is too long.
Author reply: Checked and modified accordingly.

5) Line 52: "as documented for instance" to remove.
Author reply: Done.
6) Line 56: "indeed" to remove.
Author reply: Done.

7) Line 58 :"Furthemore" to remove.
Author reply: Done.

8) Line 61: replace "important hazards" by "unexpectedly" or something like that
Author reply: Revised and modified.

9) Line 68: "T**h**ermohaline"
Author reply: Corrected.

10) Line 70: "Additionally" to remove.
Author reply: Done.

11) Line 79: "Liter" and not "Litre". I noticed this mistake several times in the manuscript.
Author reply: Thanks for noticing the error. We have corrected in all the text.

12) Line 81: "in the area" to remove.
Author reply: Done.

13) Line 83: "Meter" and not "Metre". I aslo noticed this mistake several times in the manuscript.

Author reply: Thanks for noticing the error. We have corrected in all the text.

14) Sentence line 93 to 97: "Typically…" is too long.

Checked and modified accordingly.

15) Line 111: "The steep slope", what are you talking about?

Author reply: The Patagonian continental shelf-break presents the isobaths of 100 and 200 m very close to the isobaths of 1000, 1800 and deeper (4000 to 6000 meters) in the adjacent ocean basin, which makes the profile of the continental slope very steep (see the bathymetry in figure 1).

16) Sentence line 123 to 126: "Hence,…" is too long.

Author reply: Checked and modified accordingly.

17) Sentence lines 137 to 138: "We first..." can be removed.

Author reply: Checked and modified accordingly.

18) Sentence lines 151 to 152: "In order to" can be removed. Therefore, you will need to reformulate the next sentence.

Author reply: Checked and modified accordingly.

19) Be careful to maintain consistency in your notations. For example, on line 157, you used a "-" between dates, whereas on line 158, you used "to"

Author reply: Checked and modified accordingly.

20) Similarly, units should be presented in a consistent short format throughout. For example, on line 177, change "in meter" to "m" to align with line 179, where "5 m depth" is used.

Author reply: Checked and modified accordingly.

21) Line 182: change "for 4h" by "during 4h".

Author reply: Done.

22) Line 204: a space is missing before "A1".

Author reply: Done.

23) Line 205: "Thereafter, they were…", who are "they"?

Author reply: "micrographs". Checked and modified.

24) Line 206: "Further" to remove.

Author reply: Done.

25) Sentence lines 236 to 237 "We used…" can be removed.

Author reply: We keep this sentence.

26) Line 261: "a**n** uniform…"

Author reply: Done.

27) Figure 2: figure letters a), b), c), d) are missing.

Author reply: The figure letters have been added.

28) Lines 271 to 272: "On November.." where do you see that?
Author reply: We refer to Figure 3. This was mentioned in the revised manuscript.

29) Line 309: "A reddish water discoloration…" where do you see that?
Author reply: This is an observation made by the scientists on board.

30) Figure 4: "around 60m" it is more like 50m.
Author reply: Considering the profiles at both stations the maxima is more around 60 m.

31) Line 334: "ITS", what is it?
Author reply: ITS: internal transcribed spacer. This is indicated in line 220 in Section 1.6 Genetic analysis.

32) Sentence lines 339 to 340: "This distinction..." not useful as it should be in the method section.
Author reply: This sentence is important here in the Results section and not in the M&M section, as this particular classification of the morphological taxonomic features used to identify Amphidomataceae species under light microscopy was the result of the joint effort between Dr. Guinder and Dr. Tillmann, following a detailed examination of the samples. Grouping species-specific taxonomic features based on morphological aspects was useful for cell counts under inverted microscopy.

33) Figures 6 and 7: either "scale bar = …" or "scale bar is …".
Author reply: Checked and modified accordingly.

34) Figures 2, 6 and 7: either a) or (a). Be consistent.
Author reply: Checked and modified accordingly.

35) Line 373: "Lyapunov frontal systems" means nothing. I suggest to reformulate the title of this section as "Description of the frontal systems" or something like that.
Author reply: We agree. We have modified the title as suggested.

36) Line 386: "80 cm/s" while in the figure it is in m/s.
Author reply: Checked and modified accordingly.

37) Line 408: "Moreover" to remove. And "exponent**S** (FSLE**S**)"
Author reply: Checked and modified accordingly.

38) Figure 10: modify the caption as following or something like that "FSLEs fields computed…"
Author reply: Checked and modified accordingly.

39) Lines 462-463: "And in particular" to remove.
Author reply: Checked and modified accordingly.

40) Line 486: "In fact" to remove.
Author reply: Checked and modified accordingly.

41) Line 533: "Additionally" to remove.
Author reply: Checked and modified accordingly.

42) Sentence lines 540 to 542: "In this study…", is incorrect. Retention is an in situ process and cannot be constrained by satellite observations.

Author reply: This sentence has been checked and reformulated accordingly.

43) Lines 548 to 558: Reorganize the sentences to avoid using a listing format.

Author reply: This paragraph has been moved before 4.1 section in the Discussion to highlight the key points of our findings, as suggested by the Reviewer.

---

## Author Comment (AC2)

General comments

The reviewed paper is a very good contribution that should be published, not only because it is well written, but because it very appropriately addresses the topic of the Amphidomataceae bloom, particularly since it is an offshore event in the Atlantic Ocean and has a large geographic coverage. There are many contributions on harmful blooms in coastal waters. The study not only includes field observations, but also considers the use of satellite images, oceanographic and molecular aspects and various technologies that enrich the scientific content.

This is a contribution that should be published, and with minor modifications.

Author reply: We appreciate the positive and constructive comments on our work. Below we have responded the specific comments.

Specific comments

I am struck by the title of the contribution, since it does not appear Amphidomataceae (Azadinium, Amphihdoma) and neither does the fact that the bloom in question occurs in South America, the Argentine sea shelf. Given the scarce contributions on these topics, i.e. non-coastal blooms, and in a geographic sector in which fewer scientific contributions are published, this is a fact that calls attention. It is suggested to evaluate the feasibility of changing the title considering Amphidomataceae and the geographical area in which the study was carried out.

Author reply: We have discussed the title of the paper and have certainly considered specifying 'Amphidomataceae,' 'Azaspiracids,' and the study area: the underexplored 'Patagonian Shelf in the SW Atlantic Ocean.' However, in this case, we are targeting a broader audience from the fields of physical and biological oceanography, regardless of the toxic species responsible for the bloom or the specific sea in which the event occurred. We believe the originality lies in the occurrence of a HAB with a unique nature, in terms of its magnitude, composition, toxicity, and location in ocean waters far from the coast, where mesoscale circulation plays a key role in its development and retention. Following the Reviewer's suggestion, we have incorporated the Patagonian Shelf in the title as follows: "Extraordinary bloom of toxin-producing phytoplankton enhanced by strong retention in offshore waters of the Patagonian Shelf"

Another aspect of a specific nature, but which depends both on the editorial policy of the journal in which the article is intended to be published, and on the authors' own interests, refers to the information on all the taxonomic aspects related to the Azadinium and Amphidoma taxa that are included as an appendix (Appendix A). Although it is not something substantial, I have the impression that these aspects should be considered as an independent publication, leaving in it only that which refers to the multispecies Amphidomataceae bloom.

Author reply: Many thanks for your appreciation of our effort in describing the taxonomic aspects of the dinoflagellates. We believe that presenting these detailed results not only confirms that the same bloom was captured, but also strengthens the characterization of the natural multispecificity of Amphidomataceae blooms. These dinoflagellates are small-sized cells whose taxonomy is complex and requires various methodological approaches and species-specific identification expertise. If we

were to transfer all the taxonomic information from the appendix to another publication, this one would lose valuable data that confirms we sampled the same bloom at both stations.

In the discussion, in the point 4.2., in the final part, when the authors contrast the platforms and oceanographic conditions on the Atlantic and Pacific coasts of the southern tip of South America, the information provided in the text is too limited, both from a topographic and oceanographic point of view, and aspects such as the platform characteristics, which is noticeably smaller in size, on the Pacific coast and the marked differences between the Humboldt and Malvinas currents, and the processes associated with phytoplanktonic blooms, should be included, beyond the fact that both currents originate from the Antarctic circumpolar current. These differences have implications for stability, retention, accumulation and oceanographic conditions in general, and a discussion on how these conditions can determine marked differences in the expressions of Amphidomataceae blooms on both coasts of South America, such as the records of abundance, and that these blooms occur at such notable distances from the coast, in the case ot the Atlantic secot.

Author reply: Following the Reviewer suggestion, we have briefly expanded on both Patagonian Shelves (Chilean in the Pacific and Argentinean in the Atlantic) in the revised version of the manuscript. As we discussed in Ramirez et al., 2022, https://doi.org/10.1016/j.hal.2022.102317, and Guinder et al., 2024, https://doi.org/10.1007/978-3-031-71190-9_3, Amphidomataceae is the most recently described toxin-producing group of dinoflagellates in the Argentine Sea, and similarly in Chile and Peru. Unfortunately, this group has not yet been exhaustively explored in either shelf, with even less research conducted on the Chilean shelf-break or upwelling frontal systems. In response to Reviewer #1's suggestion, we have added the reference Iriarte et al., 2023, which is a short review of the most common HABs in the Chilean coastal area – specifically fjords and channels. Here, Amphidomataceae have not yet been described as causative agents of blooms.

Other corrections

With regard to figure 2, the text (since line 261) is missing the inclusion of maximum values? or ranges of Chl-a concentrations and surface temperatures, since the figure 2 do not provide details regarding these aspects. And in the same figure, although a, b, c, and d are indicated in the legend, this detail should be included in each box of the figure.

Author reply: We agree with the reviewer that although Figure 2 has the colorbar with labels and units, both for Chl-a (in µg L-1) and for SST (in °C), maximum values and ranges are missing in the text. We have added this information in the revised version of the manuscript.

Thanks for noticing that the letters of the panels are missing. We have added them as well.

Although in general the contribution is well presented and ordered as corresponds to scientific texts, there are minor errors, particularly in the scientific names of Azadinium species, e.g. lines 342, 484 and 489

Author reply: All checked and corrected.

On the other hand, in line 536 it says Hernández-Carrazco, it should say Hernández-Carrasco.,

Author reply: Checked and corrected.

---

## Referee Report (RR1)

**General Comments**

I appreciated your further explanations regarding my questions. Concerning the structure of the Methods and Results sections, I understand why you prefer to keep the original order: the cruises were not planned to sample the extraordinary bloom, as it was unexpected, and your focus is on finding the physical explanations for your observations!

Based on your revision, I would still suggest making minor adjustments to the manuscript.

**Specific Comments**

Figures: I am still concerned about the colormap used in the figures. As the references I initially sent you mentioned:

*e.g. Michael Stoelzle and Lina Stein 2021:* *"The rainbow color map attracts attention but distorts and misleads scientific visualizations. Major rainbow pitfalls are the non-linear data encoding, steps and disorder in luminance, and minor perceptual accessibility for people with CVD or other vision impairments. Here we investigated the use of rainbow color maps in around 1000 papers in different environmental journals and found that the misleading rainbow color map or red–green color issues are present in around 44 % of all papers …"*

This is even more problematic for studies like yours, as rainbow colormaps distort fine-scale current analysis by masking gradients, amplifying irrelevant features, misrepresenting continuity, excluding colorblinds... I believe the rainbow colormap hinders rather than helps in visualizing key features. I strongly recommend using perceptual colormaps like viridis.

- Part 2.7: How do you obtain vertical velocities? Is not your model in 2D (only horizontal velocities)?

- I appreciated that you added more discussion about results of your lagrangian experiments. However the sentence line 527: "...no fronts separated…" is a bit strong, because as you mentionned later you do not investigated the submesoscale dynamics. I suggest just to add the scale you refer to: "no mesoscale front ..".

- Same as above, the sentence on line 533, "...this situation is improbable..." seems a bit too strong. Do you mean that it is improbable for peculiar conditions to independently lead to two similar blooms? If so, it is not entirely evident that this is improbable. I agree that a single bloom occurring at both stations seems more likely. However, the distinction between the 'Same bloom patch' scenario in the heterogeneous case and the 'Independent bloom' scenario is not entirely clear. Similar optimal bloom conditions could have occurred at both stations, potentially leading to either heterogeneous patches or independent blooms, as you do not have data between the both stations.

**Technical corrections**

Some of the mistakes I raised in my initial feedback still appear in the revised manuscript. For example, in the abstract, the abbreviation AZA-2 is still missing after first mention of the full name

(line 31), "metre" is still write instead of "meter" and "litre" instead of "liter". Line 73 the first "h" of thermohaline is still missing. Please refer to the previous file I sent to ensure all these corrections are made.

- Part 2.2: It would be better if the objectives of the cruises are presented before the strategies

- Line 245: You say "two complementary analyses: Lagrangian advection ...", while in the discussion you refer to both analyses as Lagrangian experiments. I would suggest using "Lagrangian analysis" to describe both analysis as you refer in the discussion.

- Lines 264 to 265 and the line 269 (from "FSLEs are commonly..") concerned the definition of FSLEs. So they should be merged together.

- Line 266: "FSLE… exponential rate ($\lambda$ [d$^{-1}$])" include the units. The equation is $\lambda(\delta) = 1/\tau(\delta) *$ $\ln(\delta_0/\delta f)$, where both $\lambda$ and $\tau$ depend on $\delta$. Also, since FSLE are defined as $\lambda$, I suggest modifying the sentence on line 266, "FSLE are strongly linked..." to something like "FSLE are defined by...".

- Lines 333 and 334: Are the two numbers after the point (31.68 and 13.69) necessary?

- Line 38: "100 km in diameter", "in" to remove

- Line 383: Which figure?

- Figure 10: "Fronts identified" – which fronts are you referring to? FSLEs identify the separation rate of a fluid. I agree that they also identify fronts (included within the Lagrangian coherent structures), but it's not entirely accurate to say it this way, especially because you are not focusing on fronts in this paper. I would suggest simply stating "FSLE ridges computed..." instead.

Is there a possibility of negative FSLE values? If so, why were they not displayed on the colorbar? Positive and negative values indicate the convergence or divergence of the flow (i.e., whether particles are getting closer or moving away)

- Figure 11: I still don't see the clear purpose of this figure. Moreover, you already have the GIF video to demonstrate this

- Lines 424-434: As I already mentioned, it's better to avoid simply listing "factors"

- Line 572: "...acted as the physical driver", change "the" by something like "a crucial" as although your study demonstrates the crucial role of mesoscale circulation in bloom formation, other physical factors could also act in synergy to create optimal bloom conditions

---

## Author Response (AR2)

**Reviewer#1**

General Comments

I appreciated your further explanations regarding my questions. Concerning the structure of the Methods and Results sections, I understand why you prefer to keep the original order: the cruises were not planned to sample the extraordinary bloom, as it was unexpected, and your focus is on finding the physical explanations for your observations!

Based on your revision, I would still suggest making minor adjustments to the manuscript.

Author reply: We are thankful for your thoughtful feedback and for understanding the reasoning behind the structure of the Methods and Results sections. We appreciate your recognition of the unexpected nature of the bloom and our focus on investigating the physical explanations behind our observations. We believe that maintaining the original order helps to highlight the context and the nature of the data collection. Thanks again for your constructive input.

Specific Comments

Figures: I am still concerned about the colormap used in the figures. As the references I initially sent you mentioned:

e.g. Michael Stoelzle and Lina Stein 2021: "The rainbow color map attracts attention but distorts and misleads scientific visualizations. Major rainbow pitfalls are the non-linear data encoding, steps and disorder in luminance, and minor perceptual accessibility for people with CVD or other vision impairments. Here we investigated the use of rainbow color maps in around 1000 papers in different environmental journals and found that the misleading rainbow color map or red–green color issues are present in around 44 % of all papers …"

This is even more problematic for studies like yours, as rainbow colormaps distort fine-scale current analysis by masking gradients, amplifying irrelevant features, misrepresenting continuity, excluding colorblinds... I believe the rainbow colormap hinders rather than helps in visualizing key features. I strongly recommend using perceptual colormaps like viridis.

Author reply: From our perspective, the features we wish to highlight are more effectively visualized using the 'jet' colormap, which we initially selected. However, we understand that this may be a matter of preference, and therefore, we do not consider it a critical issue for further discussion. In light of this, we have updated the colormaps for Figures 8, 9, and Figure D01, which display Absolute Dynamic Topography maps, from 'jet' to 'parula' in the revised manuscript. The 'parula' colormap is a sequential variant similar to 'viridis' and, in our opinion, better emphasizes the features presented in the figures

- Part 2.7: How do you obtain vertical velocities? Is not your model in 2D (only horizontal velocities)?

Author reply: We are uncertain about this comment, as we do not reference vertical velocities in Section 2.7 or elsewhere in the manuscript.

- I appreciated that you added more discussion about results of your lagrangian experiments. However the sentence line 527: "...no fronts separated…" is a bit strong, because as you mentionned later you do not investigated the submesoscale dynamics. I suggest just to add the scale you refer to: "no mesoscale front ..".

Author reply: We have modified the sentence accordingly by specifying the spatial scale, as suggested.

- Same as above, the sentence on line 533, "...this situation is improbable..." seems a bit too strong. Do you mean that it is improbable for peculiar conditions to independently lead to two similar blooms? If so, it is not entirely evident that this is improbable. I agree that a single bloom occurring at both stations seems more likely. However, the distinction between the 'Same bloom patch' scenario in the heterogeneous case and the 'Independent bloom' scenario is not entirely clear. Similar optimal bloom conditions could have occurred at both stations, potentially leading to either heterogeneous patches or independent blooms, as you do not have data between the both stations.

Author reply: We have revised this explanation to strengthen the argument that it is highly unlikely for the same phytoplankton bloom—in terms of species composition and relative abundances—to develop independently (which means that they were never connected as the heterogeneous same patch scenario) at two separate locations. Essentially, both physical and biological components play a determining role in bloom development. The revised text reads as follows:

"A less likely scenario is that (3) two Amphidomataceae blooms developed independently at both locations (Fig. 12d). This scenario is highly improbable given the complex interplay of physical and biological processes that govern bloom development. On the physical side, advection, accumulation, and stirring of water masses act as selective forces, favoring the proliferation of certain species or functional groups over others, depending on local and transient conditions (Abraham et al., 2000; Lehan et al., 2007; Della Penna et al., 2015). Biologically, additional layers of variability—including interspecific competition, grazing pressure, successional dynamics, toxin production, and cyst formation—further shape bloom composition and trajectory. Considering the influence of such dynamic and site-specific factors, the independent development of blooms with identical species composition and relative abundances at two separate locations is unlikely. Furthermore, no dormant cysts of Amphidomataceae have been reported, ruling out the possibility of localized population outbreaks from a resting stage, as has been observed for other dinoflagellates forming HABs in frontal systems (Smayda, 2002; Akselman et al., 2015)."

Technical corrections

Some of the mistakes I raised in my initial feedback still appear in the revised manuscript. For example, in the abstract, the abbreviation AZA-2 is still missing after first mention of the full name (line 31), "metre" is still write instead of "meter" and "litre" instead of "liter". Line 73 the first "h" of thermohaline is still missing. Please refer to the previous file I sent to ensure all these corrections are made.

Author reply: We have amended these mistakes in the revised version.

- Part 2.2: It would be better if the objectives of the cruises are presented before the strategies

We have modified this part according to the reviewer's suggestion.

- Line 245: You say "two complementary analyses: Lagrangian advection ...", while in the discussion you refer to both analyses as Lagrangian experiments. I would suggest using "Lagrangian analysis" to describe both analysis as you refer in the discussion.

We have kept the use of analysis and experiments as to our understand, this writing is not confusing to refer to the Lagrangian studies shown in Figs. 8, 9 and in Fig. D.

We appreciate the reviewer's attention to detail. However, we are not entirely sure we understand the suggestion. In our view, the current phrasing—"two complementary analyses: Lagrangian advection…" as well as Lagrangian experiments or Lagrangian simulations — accurately describes the approach taken and is consistent with the way the methods, results and discussion are presented. We believe the current terminology does not introduce confusion, and for this reason, we prefer to retain the original wording.

- Line 266: "FSLE… exponential rate ($\lambda$ [d $^1$])$^-$ " include the units. The equation is $\lambda(\delta) = 1/\tau(\delta) * \ln(\delta_o/\delta f)$, where both $\lambda$ and $\tau$ depend on $\delta$. Also, since FSLE are defined as $\lambda$, I suggest modifying the sentence on line 266, "FSLE are strongly linked..." to something like "FSLE are defined by...".

Corrected.

- Lines 333 and 334: Are the two numbers after the point (31.68 and 13.69) necessary?

We believe it is worth indicating these two decimal values. Including decimals allows for a more precise comparison between sampling stations, especially when dealing with high cell concentrations. For this reason, we prefer to retain the values as originally presented.

- Line 38: "100 km in diameter", "in" to remove

Corrected.

- Line 383: Which figure?

Figure 9, now added.

- Figure 10: "Fronts identified" – which fronts are you referring to? FSLEs identify the separation rate of a fluid. I agree that they also identify fronts (included within the Lagrangian coherent structures), but it's not entirely accurate to say it this way, especially because you are not focusing on fronts in this paper. I would suggest simply stating "FSLE ridges computed..." instead.

Corrected.

Is there a possibility of negative FSLE values? If so, why were they not displayed on the colorbar? Positive and negative values indicate the convergence or divergence of the flow (i.e., whether particles are getting closer or moving away)

Thank you for pointing this out. You are absolutely right—our FSLE values are indeed negative, and we inadvertently overlooked the negative sign in the colorbar. We have now corrected the color scales in Figures 10 and 11 to explicitly include the negative sign, and we have revised the corresponding section in the materials and methods and discussion sections to reflect this correction.

As you correctly mentioned, negative FSLE values indicate divergence of particles transverse to the filaments, which aligns well with our interpretation. In our case, the sampling stations are surrounded by strong negative FSLE values, suggesting that the phytoplankton bloom remains retained in place, rather than being advected away.

- Figure 11: I still don't see the clear purpose of this figure. Moreover, you already have the GIF video to demonstrate this

We consider Figure 11 important to include in the main manuscript, as it provides key information to support the daily evaluation of FSLEs and helps demonstrate that the sampling stations remained within the same water mass throughout the entire study period. While we agree that the GIF video offers a more dynamic representation of this process, it serves as a complementary digital resource. Figure 11, on the other hand, is essential for understanding and supporting the discussion within the main body of the text.

- Lines 424-434: As I already mentioned, it's better to avoid simply listing "factors"

We have decided to retain this paragraph in its current form, as we find the structure effective and the listing of key factors helpful for summarizing the main findings. In our view, this format contributes to the clarity and organization of the discussion.

- Line 572: "...acted as the physical driver", change "the" by something like "a crucial" as although your study demonstrates the crucial role of mesoscale circulation in bloom formation, other physical factors could also act in synergy to create optimal bloom conditions

Corrected.

**Reviewer#2**

As I indicated in my first review, I believe the proposed paper should be published, and in this opportunity the paper´s presentation and organization of the various aspects considered have been improved. It is well written and has many relevant details, particularly for blooms that do not occur in the coastal sector, harmful blooms that are less documented in the scientific literature.

Author reply: We sincerely thank the reviewer for the positive and encouraging feedback. We appreciate the recognition of the manuscript's improvements and the relevance of documenting less-studied, offshore harmful algal blooms.

In the introduction my suggestion when the authors refer to the negative effects of harmful blooms, that productive activities such as coastal fisheries, aquaculture, and tourism should also be included among the activities affected by these phenomena.

Author reply: We appreciate the reviewer's suggestion to explicitly mention productive activities such as coastal fisheries, aquaculture, and tourism among those affected by harmful algal blooms. As noted, the negative impacts of HABs are introduced early in the manuscript (lines 46–48), where we refer to their effects on marine biota, ecosystems, and human health. We consider that these broad categories implicitly encompass the impact on ecosystem services, including biodiversity, food quality, tourism, and aquaculture. However, we chose not to include specific examples of coastal activities in this context, as our study focuses on offshore waters, where such impacts are less direct or may differ in nature.

And in the discussion, it would be interesting if the authors could provide comments regarding the interaction of microalgal assemblages and bacterial assemblages, particularly interactions between bacterial assemblages and harmful microalgae. The authors address physical interactions, biological interactions such as predation, and chemical interactions, but there is no mention of the aforementioned aspects, given the specific nature of the blooms described.

Author reply: We appreciate the reviewer's comment highlighting the importance of interactions between microalgal and bacterial assemblages, particularly in relation to harmful microalgae. We fully agree that microbial processes, especially during the advanced and senescent stages of phytoplankton blooms, play a key role in bloom dynamics and ecosystem functioning. However, given the specific focus and scope of our study, which does not include bacterial data, we believe that incorporating a detailed discussion of this topic would go beyond the objectives of the current manuscript. For this reason, we have chosen not to expand on these interactions in the discussion section.

Finally, when metabarcoding results are delivered, it is not clear if the presented results are exclusively oriented to Azadinium and Amphidoma species, since a very low diversity is noted in the results, the fraction of sequences that could not be assigned to any specific taxon should be discussed.

Author reply: We appreciate the reviewer's comment, as we had not fully addressed this aspect regarding the genetic analysis of the samples. In response, we have now added a note in the Appendix, below the Table, with the following information, as suggested by the reviewer:

"A total of 849 ASVs were identified; however, only 118 of these were successfully annotated at the species level (with over 97% similarity). This limitation may be attributed to the inadequacy of the ITS region database used in our study, which may have affected the taxonomic resolution of certain sequences."